# Abrupt hippocampal remapping signals resolution of memory interference

Guo Wanjia 1✉, Serra E. Favila 2, Ghootae Kim3, Robert J. Molitor1 & Brice A. Kuhl 1✉

Remapping refers to a decorrelation of hippocampal representations of similar spatial environments. While it has been speculated that remapping may contribute to the resolution of episodic memory interference in humans, direct evidence is surprisingly limited. We tested this idea using high-resolution, pattern-based fMRI analyses. Here we show that activity patterns in human CA3/dentate gyrus exhibit an abrupt, temporally-specific decorrelation of highly similar memory representations that is precisely coupled with behavioral expressions of successful learning. The magnitude of this learning-related decorrelation was predicted by the amount of pattern overlap during initial stages of learning, with greater initial overlap leading to stronger decorrelation. Finally, we show that remapped activity patterns carry relatively more information about learned episodic associations compared to competing associations, further validating the learning-related significance of remapping. Collectively, these findings establish a critical link between hippocampal remapping and episodic memory interference and provide insight into why remapping occurs.

[1] Department of Psychology, University of Oregon, Eugene, OR, USA. [2] Department of Psychology, Columbia University, New York, NY, USA. [3] Korea Brain Research Institute, Dong-gu, Daegu, Republic of Korea. ✉email: wanjiag@uoregon.edu; bkuhl@uoregon.edu

The hippocampus is critical for forming long-term, episodic memories[1–3]. However, one of the fundamental challenges that the hippocampus faces is that many experiences are similar, creating the potential for memory interference[4,5]. In rodents, it is well established that minor alterations to the environment can trigger sudden changes in hippocampal activity patterns—a phenomenon termed remapping[6,7]. An appealing possibility is that hippocampal remapping also occurs in human episodic memory, allowing for similar memories to be encoded in distinct activity patterns that prevent interference[8]. At present, however, there remains an important gap between evidence of place cell remapping in the rodent hippocampus and episodic memory interference in humans. To bridge this gap, it is informative to consider how properties of place cell remapping, as demonstrated in the rodent hippocampus, might translate to episodic memory interference in humans.

One of the most important properties of remapping in the rodent hippocampus is that it is characterized by abrupt transitions between representations[9–12]. These abrupt transitions, evidenced by decorrelations in patterns of neural activity, have most typically been observed as a function of the degree of environmental change[9,11]. However, abrupt remapping can also occur as a function of experience with a new environment[10,12]. Evidence of experience-dependent remapping[6,13,14] suggests an important point: that remapping fundamentally reflects changes in internal representations, as opposed to changes in environmental states[15,16]. An emphasis on internal representations lends itself well to human episodic memory in that it suggests that hippocampal remapping should occur as memories change. More specifically, this perspective makes the critical prediction that when two events are highly similar, hippocampal remapping will occur if, and when, corresponding memories become distinct. To date, a number of human fMRI studies have observed experience-dependent decorrelations in hippocampal representations of similar memories[17–22] and/or have linked hippocampal pattern overlap to memory interference[20,23–25]. However, to test the prediction that hippocampal activity patterns abruptly remap when memory interference is resolved it is necessary to precisely track changes in memories as a function of temporally-specific changes in hippocampal representations. Critically, standard approaches of averaging fMRI data across different stimuli (memories), stimulus repetitions, and/or participants can easily obscure or wash out abrupt changes in hippocampal representations if the timing of those changes varies across memories or participants.

Evidence of place cell remapping in rodents also motivates specific predictions regarding the relative contributions of hippocampal subfields, with a major distinction being between CA3/dentate gyrus and CA1[8,26,27]. In general, CA3 and dentate gyrus are thought to be more important than CA1 for discriminating between similar stimuli[16,27–31] and remapping has been shown to occur more abruptly in CA3 than in CA1[10,12,32]. High-resolution fMRI studies in humans have also tested for and confirmed distinctions between these subfields. For example, fMRI studies have found that, relative to CA1, activity patterns/responses in CA3 and dentate gyrus are more sensitive to subtle differences between similar memories[17,19,33,34] or spatial environments[23,24,33]. Moreover, responses in the human CA3/dentate gyrus have specifically been linked to behavioral discrimination of similar memories[23,24,35]. However, these studies have not directly established a link between temporally abrupt remapping in CA3/dentate gyrus and changes in corresponding episodic memories.

Here, we tested whether the resolution of interference between highly similar episodic memories is associated with an abrupt remapping of activity patterns in the human CA3/dentate gyrus. We used an associative memory paradigm in which participants learned and were repeatedly tested on associations between scene images and object images[20]. The critical design feature was that the set of scene images included pairs of highly similar scenes (Fig. 1a). These scene pairmates were intended to elicit associative memory interference. Across six rounds of learning, we tracked improvement in associative memory for each set of pairmates while also continuously tracking representational changes indexed by fMRI. Specifically, after each associative memory test

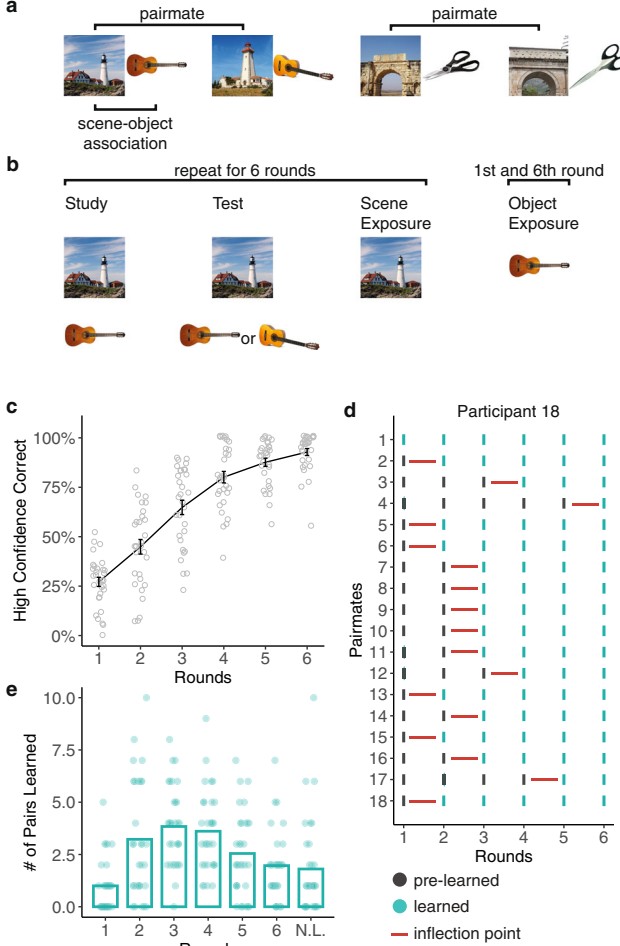

**Fig. 1 Experimental design and behavior. a** Participants learned 36 scene-object associations. The 36 scenes comprised 18 scene pairmates which consisted of highly similar image pairs (e.g., "lighthouse 1" and "lighthouse 2"). Scene pairmates were also associated with similar objects (e.g., "guitar 1" and "guitar 2"). **b** Participants completed six rounds of study, test, and exposure phases. During the study, participants viewed scenes and associated objects. During the test, participants were presented with scenes and had to select the associated object from a set of two choices, followed by a confidence rating (high or low confidence; not shown). During exposure, scenes (rounds 1–6) or objects (rounds 1 and 6) were presented and participants made an old/new judgment. fMRI data were only collected during the scene and object exposure phases. **c** Mean percentage of high confidence correct responses for each test round. **d** Data from a representative participant showing the "inflection point" in learning (red horizontal line), for each pairmate. The inflection point was defined as the point at which participants transitioned to high confidence correct retrieval for both scenes within a pairmate—a transition from "pre-learned" (black) to "learned"(aqua). **e** Mean number of scene pairmates that transition to a learned state at each round. N.L. indicates pairmates that were never learned. Notes: Data were presented as mean values ± SEM, n = 31 independent participants. Source data are provided as a Source Data file.

round, participants were shown each scene image one at a time (exposure phase) which allowed us to measure the activity pattern evoked by each scene and, critically, the representational distance between scene pairmates. To preview, we find that behavioral expressions of memory interference resolution are temporally coupled to abrupt, stimulus-specific remapping of human CA3/dentate gyrus activity patterns. This remapping specifically exaggerated the representational distance between similar memories. In additional analyses, we show that the magnitude of remapping that individual memories experienced was predicted by the degree of initial pattern overlap among CA3/dentate gyrus representations and that remapped CA3/dentate gyrus representations carried increased and highly specific information about learned episodic associations.

## Results

Participants completed six rounds of the experimental paradigm while inside an fMRI scanner. Each round included a study phase, an associative memory test phase, and a scene exposure phase (Fig. 1b). fMRI scanning was only conducted during the exposure phases. During the study phases, participants viewed scene-object associations one at a time. During the associative memory test phases, participants were shown scenes, one at a time, along with two very similar object choices (e.g., two guitars); one object was the target (i.e., the object that had been paired with the current scene) and the other object was the competitor (i.e., the object that had been paired with the scene pairmate). After selecting an object, participants indicated their confidence (high or low). During exposure phases, participants were shown each scene, along with novel scenes, and made a simple old/new judgment (mean ± 95% CI: $d' = 5.40 ± 0.88$; one-sample $t$-test vs. 0: $t_{30} = 12.58$, $p < 0.001$, Cohen's $d = 2.26$).

**Behavior**. During the associative memory test phases, participants chose the correct object with above-chance accuracy in each of the six rounds (round 1: $t_{30} = 2.65$, $p = 0.013$, $d = 0.48$, CI = [0.56 ± 0.05]; round 2: $t_{30} = 7.77$, $p < 0.001$, $d = 1.40$, CI = [0.69 ± 0.05]; round 3: $t_{30} = 10.78$, $p < 0.001$, $d = 1.94$, CI = [0.79 ± 0.05]; round 4: $t_{30} = 19.39$, $p < 0.001$, $d = 3.48$, CI = [0.87 ± 0.04]; round 5: $t_{30} = 29.71$, $p < 0.001$, $d = 5.34$, CI = [0.92 ± 0.03]; round 6: $t_{30} = 41.38$, $p < 0.001$, $d = 7.43$, CI = [0.95 ± 0.02]; one-sample $t$-tests vs. 50%). Accuracy markedly improved across rounds (main effect of round: $F_{1,30} = 318.86$, $p < 0.001$, $\eta^2 = 0.91$). The rate of choosing the correct object with high confidence also robustly increased across rounds, from a mean of 27.15 ± 4.71% in round 1 to 92.83 ± 3.58% in round 6 (main effect of round: $F_{1,30} = 574.44$, $p < 0.001$, $\eta^2 = 0.95$; Fig. 1c). See Supplementary Fig. 1 for test accuracy for each set of scene pairmates.

To test whether hippocampal remapping was temporally coupled with the resolution of memory interference, we identified, for each participant and for each set of pairmates, the learning round in which scene-object associations were recalled with high confidence (for both scenes in a pairmate). We refer to this timepoint as the "learned round" (LR; see Methods). Of critical interest for our remapping analyses was the correlation of activity patterns evoked by scene images during the LR with activity patterns evoked immediately prior to the LR-1. We refer to this transition (from pre-learned to learned) as the "inflection point" (IP) in learning (Fig. 1d). For example, if the learned run for a particular set of pairmates was round 4, then the IP was the transition from round 3 to 4. Our rationale for correlating activity patterns from the LR with activity patterns from the preceding round (LR-1) was that this correlation would capture the critical change in hippocampal representations (remapping) that putatively supports learning.

**Remapping in CA3/dentate gyrus is time-locked to the inflection point in learning**. For our fMRI analyses, our primary focus was on pattern similarity between scene pairmates. Pattern similarity was measured by correlating patterns of fMRI activity evoked by each scene during the scene exposure phases. Pairmate similarity was defined as the correlation between activity patterns evoked by scene pairmates (e.g., "lighthouse 1" and "lighthouse 2"; Fig. 2b). Correlations between scenes that were not pairmates (e.g., "lighthouse 1" and "arch 2"; Fig. 2b) provided an important baseline measure of non-pairmate similarity. We refer to the difference between these two measures (pairmate—non-pairmate similarity) as the pairmate similarity score[20]. A positive pairmate similarity score would indicate that visually similar scenes (e.g., two lighthouses) are associated with more similar neural representations than two unrelated scenes. Critically, because pairmate similarity scores are a relative measure, they can be directly compared across different brain regions[36]—something that would be inadvisable with raw correlation values. For all pattern similarity analyses, correlations were always performed across learning rounds—for example, correlating "lighthouse 1" at the LR with "lighthouse 2" at LR-1. This ensured the independence of fMRI data[37], but was also intended to capture transitions in hippocampal representations (remapping).

Following a prior study that used similar stimuli and analyses[20], fMRI analyses targeted the following regions of interest (ROIs): hippocampus, parahippocampal place area (PPA), and early visual cortex (EVC). PPA and EVC served as important control regions indexing high-level (PPA) and low-level (EVC) visual representations. We did not anticipate that these regions would demonstrate learning-related remapping. Within the hippocampus, we leveraged our high-resolution fMRI protocol to segment the hippocampus body into subfields comprising CA1 and a combined CA3/dentate gyrus (see Methods). Motivated by past empirical findings[33,38] and theoretical models[8], we predicted that remapping would occur in CA3/dentate gyrus. More specifically, we predicted that CA3/dentate gyrus remapping would occur at the IP in learning. To test this prediction, we compared pairmate similarity scores at the IP to pairmate similarity scores at a timepoint just prior to the IP (pre-IP). Whereas pairmate similarity scores at the IP were based on correlations between activity patterns from the LR and the preceding round (LR-1), pairmate similarity scores at the pre-IP were based on correlations shifted back one step in time: i.e., between LR-1 and LR-2. Thus, whereas the IP captured the transition from pre-learned to learned, the pre-IP was an important reference point that corresponded to a "non-transition" (pre-learned to pre-learned).

An ANOVA with factors of behavioral state (pre-IP, IP) and ROI (CA3/dentate gyrus, CA1, PPA, and EVC) revealed a significant main effect of ROI ($F_{3,90} = 4.08$, $p = 0.009$, $\eta^2 = 0.04$), reflecting overall differences in pairmate similarity scores across ROIs. Scores were numerically lowest in CA3/dentate gyrus and numerically highest in EVC. There was no main effect of behavioral state ($F_{1,30} = 2.71$, $p = 0.110$, $\eta^2 = 0.01$), indicating that learning did not have a global effect on representational structure across ROIs. Critically, however, the interaction between behavioral state and ROI was significant ($F_{3,90} = 2.95$, $p = 0.037$, $\eta^2 = 0.04$), indicating that learning differentially influenced pairmate similarity scores across ROIs.

Within CA3/dentate gyrus, pairmate similarity scores were significantly lower at the IP than the pre-IP ($t_{30} = -2.24$, $p = 0.033$, $d = 0.40$, CI = [−0.012 ± 0.011]), consistent with our prediction that remapping would specifically occur at the behavioral IP. Importantly, we also confirmed via permutation test (see Methods) that CA3/dentate gyrus pairmate similarity scores at the IP were lower than would be expected if the

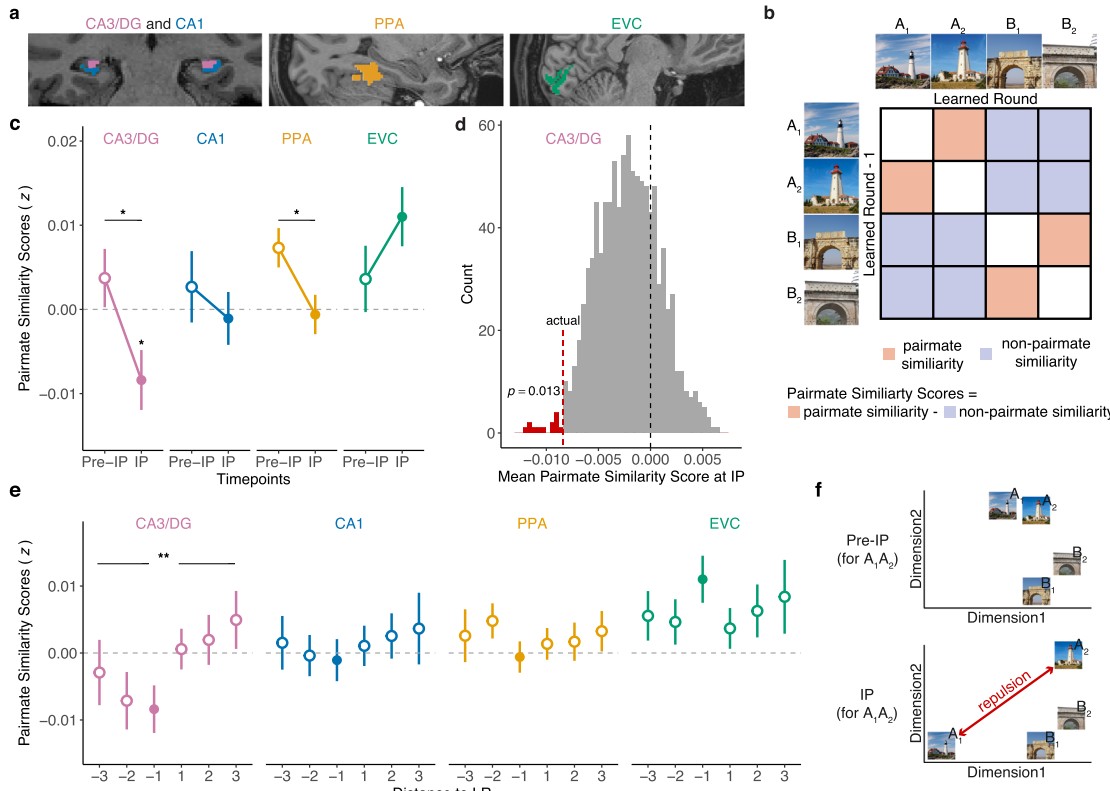

**Fig. 2 Pairmate similarity scores change at the behavioral inflection point. a** Regions of interest included CA3/dentate gyrus (CA3/DG, pink) and CA1 (blue) in the hippocampus, the parahippocampal place area (PPA, yellow), and early visual cortex (EVC, green). **b** Correlation matrix illustrating how pairmate similarity scores were computed at the behavioral inflection point. See Methods for details. **c** Pairmate similarity scores at the behavioral inflection point (IP) and just prior to the inflection point (pre-IP) across different regions of interest (ROIs). Pairmate similarity scores significantly varied by ROI ($p = 0.009$, repeated measures ANOVA) and there was a significant interaction between ROIs and behavioral state ($p = 0.037$, repeated measures ANOVA). In CA3/DG, pairmate similarity scores at the IP were significantly lower than 0 ($p = 0.025$, two-tailed one-sample $t$-test) and significantly lower than the pre-IP state ($p = 0.033$, two-tailed paired samples $t$-test). In PPA, pairmate similarity scores decreased from pre-IP to IP ($p = 0.030$, two-tailed paired samples $t$-test). **d** A permutation test (1000 iterations) was performed by shuffling, within participants, the mapping between the behavioral inflection point and scene pairmates. In CA3/dentate gyrus the actual mean group-level pairmate similarity score at the IP was lower than 98.70% of the permuted mean similarity scores ($p = 0.013$, one-tailed permutation test). **e** Pairmate similarity scores calculated by correlating the learned round (LR) with each of the three preceding rounds (– distance to LR) and each of the three succeeding rounds (+ distance to LR). [Note: the inflection point was defined as the correlation between the LR and the immediately preceding round (LR − 1); the inflection points are depicted by filled circles and are the same values as in **c**]. In CA3/dentate gyrus, pairmate similarity scores were significantly lower when the LR was correlated with preceding rounds compared to succeeding rounds ($p = 0.006$, two-tailed paired samples $t$-test). The difference was not significant for any other ROIs (CA1: $p = 0.435$; PPA: $p = 0.955$; EVC: $p = 0.760$; two-tailed paired sample $t$-tests). **f** Conceptual illustration of a decrease in pairmate similarity scores from pre-IP to IP. In the pre-IP state (top panel), $A_1$ and $A_2$ are nearby in representational space. In the IP state (bottom panel), the representational distance between $A_1$ and $A_2$ has been exaggerated. When pairmates (e.g., $A_1$ and $A_2$) are farther apart in representational space than non-pairmates (e.g., $A_1$ and $B_2$) the pairmate similarity score will be negative (i.e., pairmate similarity < non-pairmate similarity), consistent with a repulsion of competing representations. Notes: *$p <$ 0.05, **$p < 0.01$. No correction for multiple comparisons was applied given the a priori predictions for CA3/dentate gyrus. Data were presented as mean ± SEM and all data reflect $n = 31$ independent participants. Source data are provided as a Source Data file.

mapping between pairmates and IPs was shuffled within participants ($p = 0.013$, one-tailed; Fig. 2d).

Notably, CA3/dentate gyrus pairmate similarity scores not only decreased at the IP, but they were significantly below 0 at the IP ($t_{30} = -2.36$, $p = 0.025$, $d = 0.19$, CI = [−0.008 ± 0.007]). In other words, pairs of scenes with high visual similarity were represented as less similar than completely unrelated scenes in CA3/dentate gyrus. While seemingly counterintuitive, several recent fMRI studies have also found that, in certain situations, hippocampal pattern similarity is lower for similar than dissimilar events[18,20,33]. This has led to the proposal that similarity triggers a repulsion of hippocampal representations. That is, just as physical proximity triggers repulsion of like magnetic poles, representational proximity triggers repulsion of similar memories (Fig. 2f). The present results, however, provide critical evidence

that this repulsion is time-locked to—and may, in fact, underlie—the resolution of interference between competing memories.

In CA1, pairmate similarity scores did not significantly differ by learning state ($t_{30} = -0.72$, $p = 0.474$, $d = 0.13$, CI = [0.004 ± 0.01]) or differ from 0 either at the pre-IP ($t_{30} = -0.63$, $p = 0.531$, $d = 0.11$, CI = [0.003 ± 0.009]) or IP ($t_{30} = -0.34$, $p = 0.735$, $d = 0.06$, CI = [−0.001 ± 0.006]). In PPA, pairmate similarity scores decreased from pre-IP to the IP ($t_{30} = -2.28$, $p = 0.030$, $d = 0.41$, CI = [0.008 ± 0.007]), with scores significantly greater than 0 at the pre-IP ($t_{30} = 3.14$, $p = 0.004$, $d = 0.56$, CI = [0.007 ± 0.005]) but not different from 0 at the IP ($t_{30} = -0.26$, $p = 0.798$, $d = 0.05$, CI = [−0.0006 ± 0.005]). In EVC, pairmate similarity scores did not significantly vary by learning state ($t_{30} = -1.39$, $p = 0.175$, $d = 0.25$, CI = [−0.007 ± 0.01]); but there was a numerical increase from pre-IP to the IP, with scores significantly above 0 at

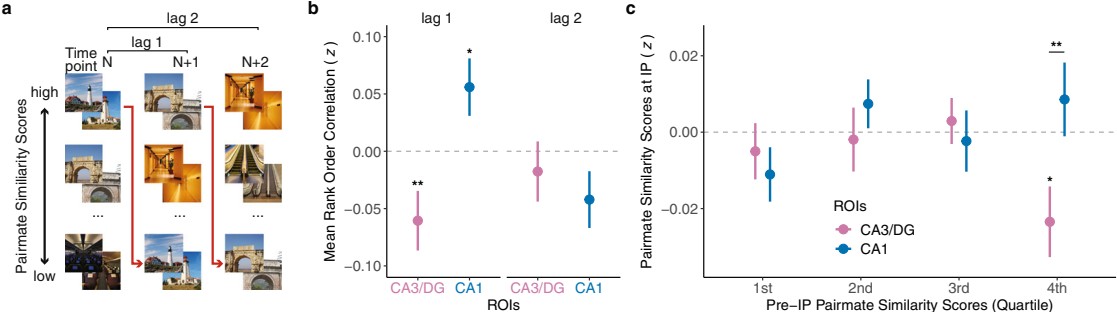

**Fig. 3 Representational structure across timepoints. a** Schematic illustration showing the rank order of scene pairmates based on pairmate similarity scores at various time points ($N$, $N + 1$, $N + 2$). If scene pairmates with relatively high pairmate similarity scores at a given timepoint are systematically associated with relatively low pairmate similarity scores at a succeeding time point (red arrows), this will produce a negative rank correlation. **b** Mean rank order correlations of pairmate similarity scores across timepoints for CA3/dentate gyrus (CA3/DG, pink) and CA1 (blue). Lag 1 correlations reflect correlations between a given timepoint and an immediate succeeding timepoint (e.g., timepoints 2 and 3). Lag 2 correlations reflect correlations between a given timepoint and a timepoint two steps away (e.g., timepoints 2 and 4). At lag 1, there was a negative correlation in CA3/dentate gyrus ($p = 0.006$, two-tailed one-sample $t$-test), but a positive correlation in CA1 ($p = 0.043$, two-tailed one-sample $t$-test). At lag 2, correlations were not significant in either CA3/dentate gyrus ($p = 0.485$, two-tailed one-sample $t$-test) or CA1 ($p = 0.120$, two-tailed one-sample $t$-test) indicating that correlations in the representational structure were specific to temporally adjacent rounds. **c** Pairmate similarity scores at the inflection point (IP) as a function of relative pairmate similarity scores in the pre-IP state (first quartile = lowest similarity, fourth quartile = highest similarity). Pairmate similarity scores in CA3/dentate gyrus were significantly lower than CA1 ($p = 0.008$, two-tailed paired samples $t$-test) and significantly below 0 ($p = 0.017$, two-tailed one-sample $t$-test) for pairmates with the highest pre-IP similarity (fourth quartile). See Supplementary Fig. 4 for the distributions of pre-IP pairmate similarity scores. Notes: \*$p < 0.05$, \*\*$p < 0.01$. No correction for multiple comparisons was applied given the a priori predictions for CA3/dentate gyrus. Data were presented as mean ± SEM and all data reflect $n = 31$ independent participants. Source data are provided as a Source Data file.

the IP ($t_{30} = 3.13$, $p = 0.004$, $d = 0.56$, CI = [0.01 ± 0.007]) but not at the pre-IP ($t_{30} = 0.92$, $p = 0.366$, $d = 0.16$, CI = [0.004 ± 0.008]).

The qualitative difference between CA3/dentate gyrus and EVC is notable in that, at the IP, these regions exhibited fully opposite representational structures: scene pairmates were more similar than non-pairmates in EVC, but less similar than non-pairmates in CA3/dentate gyrus. This finding parallels prior evidence of opposite representational structures in the hippocampus and EVC[18,20] and argues against the possibility that CA3/dentate gyrus "inherited" representational structure from early visual regions. More generally, pairmate similarity scores markedly varied across the four ROIs at the IP ($F_{3,90} = 8.73$, $p < 0.001$, $\eta^2 = 0.14$), but not at the pre-IP ($F_{3,90} = 0.33$, $p = 0.804$, $\eta^2 = 0.008$), underscoring the influence of learning on a representational structure.

For the preceding fMRI analyses, the IP was defined as the correlation between the LR and the immediately preceding round (LR-1). To more fully characterize how the representational state at the LR compared to other rounds, we additionally correlated representations at the LR to representations at LR-2 and LR-3 (i.e., other rounds that preceded the LR) and also correlated the LR with LR+1, LR+2, and LR+3 (rounds that followed the LR). Within CA3/dentate gyrus, pairmate similarity scores were significantly lower when correlating the LR with rounds that preceded learning compared to rounds that followed learning ($t_{30} = -2.98$, $p = 0.006$, $d = 0.54$, CI = [−0.009 ± 0.006]; Fig. 2e and see Supplementary Fig. 2 for related analyses). This asymmetry indicates that CA3/dentate gyrus representations expressed at the LR were systematically biased away from the initial representational position of competing memories. More generally, these data support the idea of an abrupt representational change (remapping) in CA3/dentate gyrus that was time-locked to the specific round at which learning occurred for individual pairmates. For CA1, PPA, and EVC, there were no significant differences in pairmate similarity scores when correlating the LR to rounds that preceded learning vs. followed learning (CA1: $t_{30} = -0.79$, $p = 0.435$, $d = 0.14$, CI = [−0.002 ± 0.006]; PPA: $t_{30} = 0.06$, $p = 0.955$, $d = 0.01$, CI = [−0.0002 ±

0.005]; EVC: $t_{30} = 0.31$, $p = 0.760$, $d = 0.06$, CI = [−0.001 ± 0.006]; Fig. 2e).

**Overlap of CA3/dentate gyrus representations triggers remapping.** The fact that pairmate similarity scores in CA3/dentate gyrus were negative at the IP (Fig. 2c) suggests that learning-related remapping involved an active repulsion of competing hippocampal representations (Fig. 2f). Conceptually, the key feature of a repulsion account is that separation of hippocampal representations is a reaction to initial overlap among memories[25]. Here, because we measured representational states throughout the course of learning, we were able to test this hypothesis directly. Specifically, we tested the prediction that relatively greater pairmate similarity scores (i.e., the higher overlap between memories) at a given timepoint is associated with relatively lower pairmate similarity scores (i.e., the lower overlap between memories) at a successive timepoint.

To test this hypothesis, we first translated the six learning rounds into five "timepoints" (see Methods). Each timepoint corresponded to the set of scene pair similarity scores obtained by correlating activity patterns across consecutive learning rounds [e.g., timepoint 1 = $r$(round 1, round 2)]. These scores reflected the representational structure at each timepoint (i.e., which pairmates were relatively similar, which pairmates were relatively dissimilar). We then rank correlated the pairmate similarity scores across successive timepoints [$r$(timepoint 1, timepoint 2)]. Whereas a positive rank correlation would indicate that representational structure is preserved across time points, a negative rank correlation would indicate that representational structure is inverted across time points. Critically, an inversion of representational structure is precisely what would be predicted if initial overlap among activity patterns (i.e., high pairmate similarity scores) triggers a repulsion of activity patterns (i.e., low pairmate similarity scores).

Notably, the rank correlation in CA3/dentate gyrus was significantly negative ($t_{30} = -2.99$, $p = 0.006$, $d = 0.54$, CI = [−0.06 ± 0.04], Fig. 3b). In contrast, the rank correlation in CA1

was significantly positive ($t_{30} = 2.11$, $p = 0.043$, $d = 0.38$, CI = $[0.06 \pm 0.05]$). The difference between CA3/dentate gyrus and CA1 was also significant ($t_{30} = 3.73$, $p < 0.001$, $d = 0.67$, CI = $[0.12 \pm 0.06]$). Importantly, the negative correlation in CA3/dentate gyrus cannot be explained by regression to the mean (see Methods). As a control, we also tested correlations at a lag of 2 [$r$ (timepoint $N$, timepoint $N + 2$)]; however lag 2 correlations did not significantly differ from 0 for either CA3/dentate gyrus ($t_{30} = -0.71$, $p = 0.485$, $d = 0.13$, CI = $[-0.02 \pm 0.05]$) or CA1($t_{30} = -1.60$, $p = 0.120$, $d = 0.29$, CI = $[-0.04 \pm 0.05]$). The interaction between lag (1, 2) and ROI (CA3/dentate gyrus, CA1) was also significant ($F_{1,30} = 7.09$, $p = 0.012$, $\eta^2 = 0.06$). Thus, for CA3/dentate gyrus and CA1, the representational structure at a given time point specifically predicted representational structure at a successive timepoint. Rank correlations did not differ from 0 in either PPA or EVC, either for lag 1 or lag 2 (PPA lag 1: $t_{30} = 0.83$, $p = 0.412$, $d = 0.15$, CI = $[0.02 \pm 0.05]$; PPA lag 2: $t_{30} = -0.80$, $p = 0.433$, $d = 0.14$, CI = $[-0.02 \pm 0.05]$; EVC lag 1: $t_{30} = 1.12$, $p = 0.272$, $d = 0.20$, CI = $[0.03 \pm 0.06]$; EVC lag 2: $t_{30} = 0.69$, $p = 0.493$, $d = 0.12$, CI = $[0.02 \pm 0.06]$). Additionally, rank order correlations did not differ from 0 when representational structure at timepoint $N$ was defined from EVC and representational structure at timepoint $N + 1$ (lag 1) or $N + 2$ (lag 2) was defined from CA3/dentate gyrus (lag 1: $t_{30} = -0.12$, $p = 0.902$, $d = 0.02$, CI = $[-0.003 \pm 0.05]$; lag 2: $t_{30} = -0.22$, $p = 0.825$, $d = 0.04$, CI = $[-0.005 \pm 0.05]$).

To better visualize the relationship in representational structure across successive timepoints—and to specifically connect this relationship to learning (as in Fig. 2c)—we computed pairmate similarity scores at the IP as a function of pre-IP pairmate similarity scores. Specifically, we binned all pairmates, by quartiles, according to pre-IP pairmate similarity scores, with the fourth quartile representing pairmates with the highest pre-IP pairmate similarity scores (see Methods for an additional rationale; see Supplementary Fig. 3 for alternative binning procedures). We then computed the mean pairmate similarity scores at the IP for each of the pre-IP quartiles. Again, this analysis was separately performed for CA3/dentate gyrus and CA1. An ANOVA with factors of ROI (CA3/dentate gyrus, CA1) and pairmate similarity scores at the pre-IP (four quartiles) revealed a significant interaction ($F_{3,90} = 3.19$, $p = 0.027$, $\eta^2 = 0.03$), indicating that pre-IP representational overlap was differentially related to representational overlap at the IP for CA3/dentate gyrus vs. CA1. Critically, this interaction was driven by a marked difference between CA3/dentate gyrus and CA1 when considering the bin with the highest overlap at the pre-IP (i.e., fourth quartile: $t_{30} = -2.87$, $p = 0.008$, $d = 0.51$, CI = $[-0.03 \pm 0.02]$, Fig. 3c). For CA3/dentate gyrus, pairmate similarity scores at the inflection point were significantly below 0 and numerically lowest for pairmates with the highest pre-IP similarity (fourth quartile comparison to 0: $t_{30} = -2.54$, $p = 0.017$, $d = 0.46$, CI = $[-0.023 \pm 0.019]$); the pattern in CA1 was qualitatively opposite. Collectively, these results provide theory-consistent evidence that remapping of competing representations in CA3/dentate gyrus is actively triggered by initial representational overlap.

**CA3/dentate gyrus scene representations differentiate between competing object associations**. Thus far, we have focused on similarity among neural representations evoked while viewing the scene images (scene exposure phase). However, our paradigm also included two fMRI runs during which participants viewed each of the objects associated with the scene images (object exposure phase; see Methods). This allowed us to test whether hippocampal activity patterns evoked while viewing the scenes

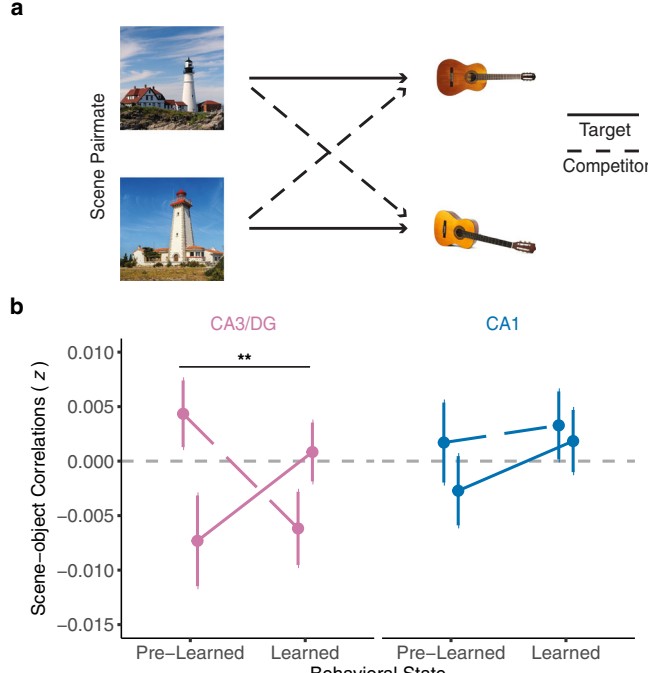

**Fig. 4 Scene-object similarity as a function of behavioral state. a** Example associations between scene pairmates and objects. The scene-object similarity was calculated by correlating activity patterns evoked during the scene exposure phases (at different behavioral states) and the object exposure phases. Target similarity refers to correlations between a given scene and the object with which it was studied. Competitor similarity refers to correlations between a given scene and the object with which its pairmate was studied. **b** Scene-object similarity as a function of object relevance (target, competitor), ROI (CA3/dentate gyrus, pink; CA1, blue), and behavioral state (pre-learned round, learned round). Mean correlations between unrelated scenes and objects (across pairmate similarity; not shown) were subtracted from target and competitor similarity values. For CA3/dentate gyrus (CA3/DG), there was a significant interaction between behavioral state and object relevance ($p = 0.002$, repeated measures ANOVA). Note: **$p < 0.01$. No correction for multiple comparisons was applied given the a priori predictions for CA3/dentate gyrus. Data were presented as mean ± SEM and all data reflect $n = 31$ independent participants. Source data are provided as a Source Data file.

resembled—or came to resemble—activity patterns evoked while viewing corresponding object images.

Whereas, pairmate similarity scores were computed by correlating activity patterns across different rounds of the scene exposure phase, here we computed correlations between a single round of the scene exposure phase and the average of the two object rounds (see Methods; see Supplementary Fig. 5 for data separated by object round). For this analysis, there were three important factors that we considered. First, we considered whether scene representations were in a "learned" state (i.e., scene representations from the LR) or a "pre-learned" state (i.e., scene representations from LR-1). Second, we separately tested correlations between each scene and (a) the target object (e.g., "guitar 1") vs. (b) the competing object (e.g., "guitar 2") (Fig. 4a). Third, we again compared results in CA3/dentate gyrus vs. CA1.

A repeated measures ANOVA with factors of ROI (CA3/dentate gyrus, CA1), behavioral state (pre-learned, learned), and object relevance (target, competitor) revealed a significant interaction between behavioral state and object relevance ($F_{1,30} = 12.42$, $p = 0.001$, $\eta^2 = 0.02$). Qualitatively, this interaction reflected a learning-related change wherein hippocampal

representations of scene images became relatively more similar to target objects and less similar to competitor objects. However, this two-way interaction between behavioral state and object relevance was qualified by a trend toward a three-way interaction between behavioral state, object relevance, and ROI ($F_{1,30} = 4.07$, $p = 0.053$, $\eta^2 = 0.01$). Specifically, the interaction between behavioral state (pre-learned, learned) and object relevance (target, competitor) was significant in CA3/dentate gyrus ($F_{1,30} = 11.98$, $p = 0.002$, $\eta^2 = 0.06$) but not in CA1 ($F_{1,30} = 0.44$, $p = 0.510$, $\eta^2 = 0.002$) (Fig. 4b). For CA3/dentate gyrus, there was a qualitative increase, from the pre-learned to learned state, in the similarity between scenes and target objects, and a qualitative decrease, from the pre-learned to learned state, in the similarity between scenes and competing objects. In other words, the remapping of CA3/dentate gyrus scene representations that occurred at the LR yielded a relative strengthening of information related to target object associations and a relative weakening of information related to competing object associations. This dissociation in CA3/dentate gyrus is notable when considering that target and competitor objects were highly similar (see Fig.1a and 4a) and even more so when considering that during the scene and object exposure phases participants were not instructed or required in any way to recall the corresponding images. The two-way interaction between behavioral state and object relevance was not significant for PPA or EVC (PPA: $F_{1,30} = 1.97$, $p = 0.170$, $\eta^2 = 0.01$; EVC: $F_{1,30} = 3.23$, $p = 0.082$, $\eta^2 = 0.02$, see Supplementary Fig. 6). Interestingly, for CA3/dentate gyrus, scene representations in the pre-learned state were significantly more similar to competitor objects than to target objects ($t_{30} = 2.70$, $p = 0.011$, $d = 0.48$, $CI = [0.012 \pm 0.009]$). While this result was not anticipated, we consider potential interpretations in the Discussion.

## Discussion

Here, we show that learning to discriminate competing episodic memories is associated with an abrupt remapping of activity patterns in the CA3/dentate gyrus. Specifically, fMRI pattern similarity in CA3/dentate gyrus decreased precisely when behavioral expressions of learning emerged. Additionally, the degree to which remapping occurred in CA3/dentate gyrus was predicted by the degree of initial pattern overlap among competing memories. Finally, remapped CA3/dentate gyrus representations contained relatively stronger information about relevant episodic associations and relatively weaker information about competing episodic associations, confirming the learning-related significance of the remapping effect.

Our experimental paradigm and analyses were inspired by—and our findings are consistent with—evidence of abrupt remapping in the rodent hippocampus[9–12]. Our findings also complement recent evidence of remapping-like phenomena in the human hippocampus[23,39,40]. However, the current findings provide unique and direct support for the proposal that hippocampal remapping is associated with the resolution of human episodic memory interference[8]. Specifically, we demonstrate an abrupt transition in hippocampal representations that occurred at an important IP in learning—the point at which participants were able to correctly discriminate similar memories and retrieve associations with high confidence. Notably, this finding was only possible because (a) we repeatedly probed episodic memory and hippocampal representations over the course of learning and (b) we identified IPs in a participant- and pairmate-specific manner. Indeed, IPs varied considerably across and within participants (Fig. 1d, Supplementary Table 1, and Supplementary Fig. 1) and the observed hippocampal remapping effect was significantly weaker when the specific mapping between behavior and fMRI data was shuffled within participants (Fig. 2d).

The fact that CA3/dentate gyrus remapping occurred precisely at the IP in learning strongly suggests that remapping was related to learning. This argument is also reinforced by our independent finding that remapped CA3/dentate gyrus activity patterns, evoked while participants viewed individual scene images, carried more information (compared to the pre-learning state) about target vs. competing object associations. In other words, the IP defined from behavioral expressions of associative memory also captured a critical change in associative representations encoded in CA3/dentate gyrus activity patterns. The fact that CA3/dentate gyrus exaggerated the representational distance between competing scenes (remapping) while simultaneously reflecting learned associations (scene-object similarity) is consistent with the idea that CA3 balances both pattern separation and pattern completion mechanisms[4,27,28,41]. The fact that remapped activity patterns contained information about learned associations is also consistent with the argument that hippocampal remapping does not simply reflect changes in the external environment—which did not change over the course of the experiment—but instead fundamentally reflects changes in internal models of the environment[15,16].

One aspect of our findings which does not, to our knowledge, have a direct analog in rodent studies of remapping is the negative pairmate similarity score we observed at the IP in CA3/dentate gyrus. The negative score indicates that scene pairmates—which were highly similar images—were associated with less overlapping CA3/dentate gyrus representations than completely unrelated scenes. In rodents, the most extreme version of remapping occurs when two similar environments are associated with fully independent place codes[8]. In our study, however, if each scene was associated with an independent representation, then the similarity between pairmates would be equal to, but not lower than, the similarity between non-pairmates. Instead, the negative pairmate similarity score requires a dependence between competing hippocampal representations wherein a given memory representation systematically moves away from the representational position of a competing memory (Fig. 2f). We refer to this dependence as "repulsion" in order to emphasize the oppositional influence that competing memories exerted. Several recent human fMRI studies have reported conceptually similar effects in the hippocampus[18,20,22,42]—and in CA3/dentate gyrus, specifically[17,19,24,33]. However, the current findings directly establish that the repulsion of competing hippocampal representations is temporally coupled to the resolution of memory interference.

Based on computational models[25,43,44], our prediction was that the repulsion effect in CA3/dentate gyrus was a direct consequence of initial overlap among activity patterns. Indeed, a recent study found that hippocampal repulsion was more likely to occur for behaviorally confusable memories[18], potentially because confusable memories are associated with greater pattern overlap during initial learning. In the current study, we tested—and confirmed—this account directly. Specifically, we found that the representational structure (relative pairmate similarity) in CA3/dentate gyrus at a given timepoint was negatively correlated with representational structure at an immediately following timepoint. This negative relationship is highly consistent with the idea that overlap, itself, triggers plasticity that "punishes" those features which are shared across memories[19,25,43,44]. While our study does not afford inferences about the causal relationship between repulsion and learning, the idea that repulsion (or remapping more generally) is triggered by representational overlap, combined with the fact that remapping was associated with learning, is consistent with the possibility that repulsion of CA3/dentate gyrus representations is a causal factor in learning. This account also offers a potential explanation for an otherwise surprising

finding: that CA3/dentate gyrus scene representations from the round that immediately preceded learning (LR-1) were significantly more similar to competitor objects than to target objects. Although speculative, it is possible that when a given scene activated the "wrong" object association (at LR-1), this actively triggered a correction in favor of the target object association that supported learning. This account is consistent with evidence that prediction errors can powerfully drive episodic memory[45,46] as well as differentiation of hippocampal activity patterns[19]. More broadly—and consistent with our findings, in general—prediction errors may induce abrupt state changes in the hippocampus that facilitate the separation of episodic memories[47].

Across multiple analyses, we observed dissociations between CA3/dentate gyrus and CA1. The fact that the remapping effects were selective to CA3/dentate gyrus is consistent with evidence from rodent studies of remapping and pattern separation[8,26,28] and with several human fMRI studies[17,19,24,28,33]. Perhaps the most notable dissociation between CA3/dentate gyrus and CA1 comes from our analysis of representational structure across time points. Whereas CA3/dentate gyrus exhibited a negative rank correlation across successive timepoints, CA1 exhibited a positive rank correlation (Fig. 3b). Thus, in contrast to CA3/dentate gyrus, CA1 was characterized by stability (though only modest stability) of representational structure across timepoints[4]. This dissociation between CA3/dentate gyrus and CA1 is consistent with the idea that CA3, in particular, supports rapid plasticity that allows for changes in memory representations on short time scales[48] and is also consistent with the evidence of faster remapping in CA3/dentate gyrus than in CA1[10,12,32]. It is also notable that the remapping effect we observed in CA3/dentate gyrus at the IP in learning strongly contrasted with the pattern of data in EVC. Whereas CA3/dentate gyrus exhibited a negative pairmate similarity score at the IP, EVC exhibited a significant, positive pairmate similarity score at the IP. This finding makes the important point that CA3/dentate gyrus was not inheriting representational structure from early sensory regions (e.g., due to visual attention) —rather, CA3/dentate gyrus fully inverted the representational structure that was expressed in EVC[20].

Taken together, our findings reveal remapping of human CA3/dentate gyrus representations that is temporally coupled to the resolution of episodic memory interference. These findings were motivated by—and complement—existing evidence of remapping in the rodent hippocampus. Yet, our findings also go beyond existing rodent or human studies by establishing a direct link between remapping and changes in internal memory states[15,16]. Additionally, our conclusion that overlap among CA3/dentate gyrus representations actively triggers a repulsion of memory representations has important implications for theoretical accounts of how the hippocampus resolves memory interference[5,8,28,43] and will hopefully inspire targeted new analyses that test for similar mechanisms in rodent models.

## Methods

**Participants**. Thirty-six participants (21 female; mean age = 23.69 years, range = 18–34 years) were enrolled in the experiment following procedures approved by the University of Oregon Institutional Review Board. Written informed consent was collected for each participant prior to the experiment. All participants were right-handed native-English speakers with normal or corrected-to-normal vision, with no self-reported psychiatric or neurological disease. One participant was excluded due to excess motion in the scanner (max FD >3.5 mm); another four participants were excluded due to low behavioral performance (see Results for more details). The final analysis included 31 participants. All participants received monetary compensation for participating.

**Stimuli**. Thirty-six images of scenes and 36 images of everyday objects were used in the experiment. The set of 36 scenes and the set of 36 objects were each comprised of 18 "pairmates" of visually and semantically similar images (Fig. 1a). An

additional 36 scenes and 12 objects were used as lures for the scene and object exposure phases of the study, respectively. Separately for each participant, scene pairmates were randomly assigned to object pairmates (Fig. 1a). For example, if "lighthouse 1" was assigned to "guitar 1", then "lighthouse 2" would be assigned to "guitar 2." Note: the scene and object images shown in the figures are not the actual stimuli used in the experiment, but are public domain images representative of the stimuli that were used. See Data Availability for access to the actual stimuli.

**Experimental procedure**. After providing consent and reviewing the instructions, participants entered the MRI scanner. Inside the scanner, participants completed six rounds of the experimental paradigm (Fig. 1b). The first round and the last round included four phases: study, test, scene exposure (scanned), and object exposure (scanned). Rounds 2–5 were the same, except they did not include the object exposure phase. Across all phases, stimuli were displayed on a gray background, projected from the back of the scanner. After exiting the scanner, participants completed a separate memory task that involved learning new scene-object associations (not reported here). The experiment was implemented in PsychoPy[49] and lasted ~3 h, with about 2 h 15 min inside the scanner.

*Study Phase*. During the study phases, participants learned 36 scene-object associations, one association at a time. Each trial began with the presentation of a scene image (1000 ms), followed by a white fixation cross (200 ms), the associated object image (1000 ms), and then another white fixation cross (1200 ms) until the start of the next trial. The order in which the 36 scene-object associations were studied was randomized for each round and for each participant.

*Test Phase*. During the test phases, participants attempted to retrieve the object associated with each of the 36 scenes. Each trial began with the presentation of a scene (1000 ms), followed by a white fixation cross (200 ms), and then the presentation of two object pairmates (e.g., "Guitar 1" and "Guitar 2"). One of the object images was the "target" (i.e., the object associated with the cued scene) and the other object image was the "competitor" (i.e., the object associated with the cued scene's pairmate). Participants had a maximum of 4000 ms to select the correct object image (target) via a button box in their right hand. If no response was made, the next trial began after a white fixation cross was displayed for 1200 ms. If a response was made, a confidence rating then appeared beneath the objects and participants had a maximum of 3000 ms to indicate whether their response was a "Guess" or "Sure." After indicating their confidence (or after time ran out), a white fixation cross appeared (1200 ms) until the start of the next trial. The location of the correct object (left or right) and the order in which each of the 36 scene-object associations were tested were randomized for each round and for each participant.

*Scene exposure phase*. During the scene exposure phases, which were conducted during fMRI scanning, participants saw 39 scene images in each of two blocks (78 scenes per round). Each block included the 36 studied scenes and three novel lure scenes. Participants made an old/new judgment for each scene. Each trial began with the presentation of a scene image (500 ms), followed by a red fixation cross (1500 ms) which represented the response window. Participants again responded using the button box. After the red fixation cross, a white fixation cross (2000 ms) was presented until the start of the next trial. The order of the 39 scene trials within each block was randomized for each block, round, and participant. Between the two blocks of 39 trials, participants performed a short odd/even judgment task (four trials). Each odd/even trial consisted of a single-digit number displayed on the screen (500 ms), followed by a red fixation cross (1000 ms) which represented the response window, and then a white fixation cross (1000 ms) until the start of the next trial.

*Object exposure phase*. The object exposure phase (conducted during fMRI scanning) was only included in the first and sixth rounds and followed an identical structure and procedure as the scene exposure phase. The only difference was that the 39 trials in each block corresponded to the 36 studied objects and three novel lure objects.

**MRI acquisition**. All images were acquired on a Siemens 3 T Skyra MRI system in the Lewis Center for Neuroimaging at the University of Oregon. Functional data were acquired with a T2*-weighted echo-planar imaging sequence with partial-brain coverage that prioritized full coverage of the hippocampus and EVC (repetition time = 2000 ms, echo time = 36 ms, flip angle = 90°, 72 slices, 1.7 × 1.7 × 1.7 mm voxels). A total of eight functional scans were acquired. Each functional scan comprised 177 volumes and included 10 s of lead-in time and 10 s of lead-out time at the beginning and end of each scan, respectively. The eight functional scans corresponded to six scans of the scene exposure phase (scans 1 and 3–7) and two scans of the object exposure phase (scans 2 and 8). Anatomical scans included a whole-brain high-resolution T1-weighted magnetization prepared rapid acquisition gradient-echo anatomical volume (1 × 1 × 1 mm voxels) and a high-resolution (coronal direction) T2-weighted scan (0.43 × 0.43 × 2 mm voxels) to facilitate segmentation of hippocampal subfields.

**Anatomical data preprocessing**. Preprocessing was performed in Python 3.7 using *fMRIPrep* 1.5.0[50,51] (RRID:SCR_016216), which is based on *Nipype* 1.2.2[52,53] (RRID:SCR_002502). The T1-weighted (T1w) image was corrected for intensity nonuniformity (INU) with N4BiasFieldCorrection[54] (ANTs 2.2.0[55], RRID: SCR_004757), and used as the T1w reference throughout the workflow. The T1w reference was skull-stripped with the antsBrainExtraction.sh workflow (ANTs) in *Nipype*, using OASIS30ANTs as the target template. Brain tissue segmentation of cerebrospinal fluid (CSF), white-matter (WM), and gray-matter (GM) was performed on the brain-extracted T1w using fast[56] (FSL 5.0.9, RRID:SCR_002823). Volume-based spatial normalization to one standard space (MNI152NLin2009-cAsym) was performed through nonlinear registration with antsRegistration (ANTs 2.2.0), using brain-extracted versions of both T1w reference and the T1w template. ICBM 152 Nonlinear Asymmetrical template version 2009c[57] (RRID: SCR_008796; TemplateFlow ID: MNI152NLin2009cAsym) was used for spatial normalization.

**Functional data preprocessing**. For each of the eight BOLD scans per participant, the following preprocessing was performed. First, a reference volume and its skull-stripped version were generated using *fMRIPrep*. A deformation field to correct for susceptibility distortions was estimated based on two echo-planar imaging (EPI) references with opposing phase-encoding directions, using 3dQwarp, AFNI[58]. Based on the estimated susceptibility distortion, an unwarped BOLD reference was calculated for a more accurate co-registration with the anatomical reference. The BOLD reference was then co-registered to the T1w reference using bbregister (FreeSurfer) which implements boundary-based registration[59]. Co-registration was configured with six degrees of freedom. Head-motion parameters with respect to the BOLD reference (transformation matrices and six corresponding rotation and translation parameters) were estimated before any spatiotemporal filtering using mcflirt FSL 5.0.9[60]. BOLD scans were slice-time corrected using 3dTshift AFNI[58] (RRID:SCR_005927). The BOLD time-series (including slice-timing correction when applied) were resampled onto their original, native space by applying a single, composite transform to correct for head-motion and susceptibility distortions. Framewise displacement (FD) confounding time-series were calculated based on the resampled BOLD time-series for each functional scan[61].

**fMRI first-level general linear model (GLM) analyses**. After *fMRIPrep* preprocessing, the first five volumes (10 s) of each functional scan were discarded. Then, the brain mask generated by *fMRIPrep* from the T1 anatomical image was used to perform brain extraction for each of the eight functional scans. Each functional scan was then median centered. For the six scans of the scene exposure phase and two scans of the object exposure phase, all first-level GLMs were performed in participants' native space with *FSL* using a Double-Gamma HRF with temporal derivatives, implemented with *Nipype*. GLMs were calculated using a variation of the Least Squares—Separate method[62]: a separate GLM was calculated for each of the 36 scenes (for scene exposure phases) or objects (for object exposure phases) across both repeats within a scan. For each GLM, there was one regressor of interest (representing a single scene or object image across its two repetitions per scan). All other trials (including lure images), FD, xyz translation, and xyz rotation were represented with nuisance regressors. Additionally, a high pass filter (128 Hz) was applied for each GLM. This model resulted in 36 beta-maps per scan (one map per scene/object) which were converted to *t*-maps that represented the pattern of activity elicited by each scene/object for each scan.

**Regions of interest**. A region of interest (ROI) for EVC was created from the probabilistic maps of Visual Topography[63] in the MNI space with a 0.5 threshold. This ROI was transformed into each participant's native space using inverse T1w-to-MNI nonlinear transformation. For each participant, the top 300 EVC voxels were then selected by averaging the *t*-maps of all scenes and objects and then choosing the voxels with the highest *t*-statistics (i.e., the voxels most responsive to visual stimuli). An ROI for the PPA was created by first using an automated meta-analysis in Neurosynth with the key term "place". Then, clusters were created using voxels with a *z*-score >2 based on the Neurosynth associative tests. Since these clusters were generated through an automated meta-analysis and were not anatomically exclusive to PPA, we visually inspected the results and manually selected the two largest clusters that were spatially consistent with PPA. One cluster was in the right hemisphere (voxel size = 247) and one cluster was in the left hemisphere (voxel size = 163). These clusters were combined into a single PPA mask. This mask was then transformed into each participant's native space using the inverse T1w-to-MNI transformation. For each participant, a final PPA ROI was generated by averaging the *t*-maps of all scene exposure phase scans and then selecting the 300 voxels with the highest average *t*-statistics (i.e., the most scene-responsive voxels). To create hippocampal ROIs, we used the Automatic Segmentation of Hippocampal Subfields (ASHS)[64] toolbox with the upenn2017 atlas to generate subfield ROIs in each participant's hippocampal body, including CA3/dentate gyrus (which included CA2, CA3, and dentate gyrus) and CA1. The most anterior and posterior slices of the hippocampal body were manually determined for each participant based on the T2-weighted anatomical structure (see Supplementary Fig. 7 for a sample demarcation). Each participant's subfield segmentations were also manually inspected to ensure the accuracy of the segmentation protocol. Then,

each subfield ROI was transformed into each participant's native space using the T2-to-T1w transformation, calculated with FLIRT (fsl) with six degrees of freedom, implemented with *Nipype*. All ROIs were again visually inspected following the transformation to native space to ensure the ROIs were anatomically correct.

**fMRI pattern similarity analyses**

*Pairmate similarity scores.* Pattern similarity was calculated as the Fisher *z*-transformed Pearson correlation between *t*-maps within each ROI. All pattern similarity analyses were performed by correlating the *t*-maps for stimuli across scans (i.e., correlations were never performed within the same scan). For our primary analyses related to pattern similarity between scene images, of critical interest was the similarity between pairmate scenes (pairmate similarity) relative to the similarity between non-pairmate scenes (non-pairmate similarity). Specifically, for each set of pairmates, the mean non-pairmate similarity was subtracted from mean pairmate similarity to yield a pairmate similarity score for each set of pairmates. As an example, to calculate pairmate similarity scores for "lighthouse 1" and "lighthouse 2" across scans 3 and 4, pairmate similarity would be defined as the mean of the following two *z*-transformed correlations: $r(\text{lighthouse } 1_{\text{scan } 3}, \text{lighthouse } 2_{\text{scan } 4})$ and $r(\text{lighthouse } 2_{\text{scan } 3}, \text{lighthouse } 1_{\text{scan } 4})$. Corresponding non-pairmate similarity scores would be defined as the mean of all *z*-transformed correlations, across the same scans (scans 3 and 4), between either pairmate (lighthouse 1 or lighthouse 2) and each non-pairmate stimulus [e.g., $r(\text{lighthouse } 1_{\text{scan } 3}, \text{arch } 1_{\text{scan } 4})$, $r(\text{arch } 2_{\text{scan } 3}, \text{lighthouse } 1_{\text{scan } 4})$, …].

*Learned round.* To relate pairmate similarity scores to behavioral measures of learning, we identified the LR for each pairmate, separately for each participant. The LR was based on performance in the associative memory test. Specifically, the LR was defined as the first round in which the target object was selected with high confidence for both scenes in a pairmate, with the additional requirement that performance remained stable in all subsequent rounds. It was, therefore, possible that both scenes in a pairmate were associated with high confidence correct responses in round N, not in round $N + 1$, and then (again) in round $N + 2$ and thereafter; in this case, the LR would be round $N + 2$.

*Inflection point.* The IP was defined as the transition from LR-1 (the round that immediately preceded the LR) to LR (the learned round). Thus, pairmate similarity scores at the IP were based on correlations of *t*-maps from LR-1 with *t*-maps from LR. We hypothesized that the behavioral state change from LR-1 to LR would correspond to a reduction in pairmate similarity scores. Pairmate similarity scores at the IP were contrasted against the "pre-IP" state, which was based on the correlation of *t*-maps from LR-2 and LR-1 (i.e., a non-transition from "pre-learned" to "pre-learned") (Fig. 2c). Pairmates for which participants never reached and sustained high confidence correct responses (mean ± s.d., 1.81 ± 2.27 per participant) and pairmates that were learned in the first round (LR = 1; mean ± s.d., 1.00 ± 1.26) were excluded from the IP analyses because neither the pre-IP nor IP states could be measured. For pairmates that were learned in the second round (LR = 2; mean ± s.d., 3.23 ± 2.80), pattern similarity at the IP was calculated and included in the analyses, but pattern similarity at the pre-IP state could not be calculated because an LR-2 did not exist. For the rest of the pairmates (LR = 3, 4, 5, or 6), we calculated pattern similarity for both the pre-IP and IP states (Fig. 1e). Similar restrictions applied to correlations between LR and LR-3, LR + 1, LR + 2, and LR + 3 (Fig. 2e). The number of pairmates included in each comparison and for each participant are reported in Supplementary Table 1.

*Representational structure across time points.* To test whether representational overlap triggered remapping (related to Fig. 3), the six rounds were translated into five timepoints. Each timepoint corresponded to a pair of consecutive rounds ([1,2], [2,3], [3,4], [4,5], and [5,6]). For each timepoint, pairmate similarity scores were calculated, as described above, by correlating activity patterns from consecutive rounds (e.g., pairmate similarity scores at timepoint 1 were based on correlations between round 1 and round 2). This yielded a set of pairmate similarity scores at each of the five timepoints. These sets of similarity scores reflected the representational structure at each timepoint (i.e., which pairmates were relatively similar and which pairmates were relatively dissimilar). Pairmate similarity scores were then correlated across timepoints using Spearman's rank correlation (Fisher *z*-transformed). Lag 1 correlations refer to rank correlations between successive timepoints whereas lag 2 correlations refer to correlations between timepoints two steps apart. To facilitate a direct comparison between lag 1 vs. lag 2 correlations, correlations were computed for the following timepoints: Lag 1 = $r$ (timepoint 1, 2), $r$(timepoint 2, 3), and $r$(timepoint 3, 4); Lag 2 = $r$(timepoint 1, 3), $r$(timepoint 2, 4), and $r$(timepoint 3, 5). It is important to emphasize that we did not correlate initial pairmate similarity scores with the change in pairmate similarity as this would produce an artifactual correlation (via regression to the mean). In contrast, a negative rank correlation (as we observed in CA3/dentate gyrus) cannot be explained by regression to the mean. Mathematically, if all values at timepoint N partially regressed toward the mean at timepoint $N + 1$, this would yield a positive rank correlation (i.e., the representational structure would be partially preserved). If all values fully regressed toward the mean (i.e., variance at timepoint $N + 1 = 0$), this would yield a null correlation ($r = 0$; representational structure fully abolished).

To specifically consider the relationship between representational structure at pre-IP and representational structure at the IP (related to Fig. 3c), we binned pairmates, by quartile (using the *cut* function in base R), according to pairmate similarity scores at pre-IP and then computed pairmate similarity scores, for each quartile, at the IP. The quartile analysis was performed within subject and separately for CA3/dentate gyrus and CA1. The mean number of pairmates included in each pre-IP bin were 3.42, 2.90, 3.10, and 2.55 for quartiles 1–4, respectively. The decision to divide pre-IP pairmate similarity scores into four bins was motivated by evidence, from conceptually related studies, of non-monotonic relationships between initial memory activation/competition and experience-dependent plasticity[43,65]. While formally testing for non-monotonic relationships was beyond the scope of the current study, the goal was to allow for qualitative inspection of the relationship. Notably, similar results were obtained when pre-IP pairmate similarity scores were binned by terciles or quintiles (Supplementary Fig. 3).

*Scene-object similarity*. To calculate pattern similarity between scenes and objects (related to Fig. 4), activation patterns for objects were first generated by averaging *t*-maps across the two object exposure phases, resulting in a single, mean activity pattern for each object. These object-specific activity patterns were then correlated with activity patterns from the scene exposure phases at LR-1 (i.e., the pre-learned state) and LR (i.e., the learned state). Correlations were separated into three groups: (1) target correlations refer to the correlation between a scene and the object it was associated with during the study phase (e.g., "lighthouse 1" and "guitar 1"), (2) competitor correlations refer to the correlation between a scene and the object that was associated with that scene's pairmate during the study phase (e.g., "lighthouse 1" and "guitar 2"), and (3) across pairmate correlations refer to correlations between a scene and an object that was not associated with that scene or its pairmate during the study phase (e.g., "lighthouse 1" and "scissors 1"). Target and competitor correlations were expressed relative to across pairmate correlations.

**Statistics and reproducibility**. To compare pairmate similarity scores and other measures across ROIs and learning states, repeated measures ANOVAs and paired samples *t*-tests were used. To test whether pairmate similarity scores and other measures were significantly positive or negative (i.e., above/below 0), one-sample *t*-tests were used. To test whether the negative pairmate similarity score observed in CA3/dentate gyrus at the IP depending on the specific mapping between behavioral and fMRI measures, we randomly shuffled the mapping between the behavioral IP and scene pairmate, within each participant (see Fig. 1d), and then computed the group-level mean pairmate similarity score at the permuted IP. This was repeated 1000 times, producing a distribution of 1000 permuted means. The observed pairmate similarity score at the IP was then compared against this distribution of permuted means. Data analysis was performed in R 3.5.0 and its associated libraries. All of the data and results reported here reflect a single experiment; an independent replication was not conducted.

**Reporting Summary**. Further information on research design is available in the Nature Research Reporting Summary linked to this article.

## Data availability
The MRI data generated in this study have been deposited on Openneuro.org (DOI: 10.18112/openneuro.ds003707.v1.0.0)[66]. The stimuli used and the behavioral data generated in this study have been deposited on osf.io (https://doi.org/10.17605/OSF.IO/VPQ2X)[67]. The source data underlying all Figures and Supplementary Figures are provided as a source data file with this paper. Source data are provided with this paper.

## Code availability
Analysis scripts are available at [https://github.com/wanjiag/NEUDIF_analysis].

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

## Acknowledgements

This work was supported by NIH-NINDS R01 NS089729 awarded to B.A.K. and NIH-NEI K00EY031607 awarded to S.E.F. Photographs used in the Figures were either released into the public domain without conditions or, for the following photographs, were reproduced with minimal edits (cropping, resizing, and/or rotating) with released right to share and remix under the following licenses (photographs are listed in order of first appearance, from left-to-right, top-to-bottom): Lighthouse 1 (Figs. 1a, b, 2b, f, 3a, and 4a): Rapidfire, CC BY-SA 3.0 https://creativecommons.org/licenses/by-sa/3.0; Lighthouse 2 (Figs. 1a, 2b, f, 3a, and 4a): H. Zell, CC BY-SA 3.0 https://creativecommons.org/licenses/by-sa/3.0; Arch 1 (Figs. 1a, 2b, f, and 3a): High Contrast, CC BY 3.0 DE https://creativecommons.org/licenses/by/3.0/de/deed.en; Escalator 1 (Fig. 3a): Basile Morin, CC BY-SA 4.0 https://creativecommons.org/licenses/by-sa/4.0; Escalator 2 (Fig. 3a): Gordon Joly, CC BY-SA 3.0 https://creativecommons.org/licenses/by-sa/3.0; Scissors 2 (Fig. 1a): Crisco 1492, CC BY-SA 4.0 https://creativecommons.org/licenses/by-sa/4.0.

## Author contributions

G.W., G.K. and B.A.K. designed the experiment. G.W. and B.A.K. analyzed the data. S.E.F. and R.J.M. consulted on data analyses. G.W. and B.A.K. wrote an initial draft of the manuscript and incorporated edits and comments from G.K., S.E.F. and R.J.M.

## Competing interests

The authors declare no competing interests.

## Additional information

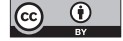

