## [Peer Review File · Nature Communications]

REVIEWER COMMENTS

Reviewer #1 (Remarks to the Author):

Guo and colleagues present novel fMRI results testing the notion that behavioral resolution of interference among similar memories is supported by hippocampal pattern repulsion or remapping. Specifically, patterns in the CA23DG subregion were found to remap as each participant successfully encoded scene-object pairs and maintained the correct representation for the rest of the experiment, without confusing it with competitor stimuli. This reconfiguration appears to be particularly strong for stimuli with high initial overlap. The data are novel and compelling, and the close link between behavioral markers of interference resolution and neural patterns is especially important and interesting. The manuscript is well-written and the statistical analyses are appropriate and described clearly. I do, however, have a few comments and questions which I hope the authors can address.

1. The data in Figure 2C clearly suggest that pairmate similarity decreases from pre-IP to IP in CA23DG. However, I'm having a bit of difficulty understanding how these data relate to Fig 2E. If I'm interpreting panel E correctly, these data suggest that pre-inflection point (LR-1), the pairmate representations are dissimilar to the learned round representation. After the inflection point (LR+1 onwards), the pairmates then actually become more similar, and remain more similar. At the level of interpretation, would this suggest that the pairmate representations actually become more integrated post-IP when they are behaviorally more differentiated?

Perhaps one way to address this question would be to compare pre-IP to post-IP representations. If the representations are repelled at the inflection point, one might imagine that the demonstrated sharp drop from pre-IP to IP should be sustained. This would be reflected in a similar magnitude of difference between pre-IP to post-IP as pre-IP to IP.

Alternatively, if I misunderstood the analysis, more detail and description of the pattern of results in panel E would be welcome here to avoid confusion.

2. The link between the behavioral inflection point and neural pattern differentiation is key to interpreting these results. However, it would also be valuable to plot neural differentiation compared to pre-learning over time, similar to the behavioral plot in Fig 1c. Most participants reach the inflection point for most stimuli by the 3rd round of the experiment – do these representations continue drifting further from the start?

3. Related to the point above, the rank order correlation analysis is interesting and supports the interpretation that representational structure is negatively related across successive timepoints. However, it would be informative to see the successive rank N+1 correlation values plotted across time, rather than averaged per lag. Fig 3 shows lag 1 vs. lag 2, but not the change in similarity over time. It would be helpful to see lag 1 and lag 2 values split by round (where possible).

4. The finding that pairmates with the highest pre-IP similarity became more differentiated is very interesting. This analysis is thoughtful and gets at the issue of what's driving the observed remapping. However, it would also be helpful to see a plot of the distribution of all the pre-IP pairmate similarity values (i.e., how much did the pre-IP values for each quartile differ across participants relative to their IP representations).

Further, were the quartiles calculated within each participant and each subregion? The calculation approach should be included in the Methods.

5. The results in Fig 4 are compelling and suggest that learning reconfigured object-scene associations as well. However, it's not immediately clear to me why the object exposure scans were averaged across the first and final rounds. Since the two object rounds were in the first and final blocks, they could also be classified as pre- and post-learning. The split of object representations into early and late scans seems even more important given the interaction with behavioral state. Further, do object representations also show any reconfiguration from pre- to post-learning, akin to scenes? If both first- and final-round object representations show the same pattern, this does not necessarily change the interpretation. However, it could suggest something interesting about the nature of these associations, such that scenes can reconfigure object representations, but not vice versa.

6. On p. 13, the authors state that the dissociation in representations across brain regions suggests that CA23DG did not simply inherit the representational structure from EVC. The data support this view, but I found this to be an interesting issue that could be tested even more directly - specifically, is higher overlap between memories in EVC in any way related to repulsion (or integration) in CA23DG?

Reviewer #2 (Remarks to the Author):

Guo and colleagues present timely and solid work on remapping in human hippocampal subfields CA3 and dentate gyrus and its role in the resolution of memory interference.

In my opinion, the following strengths stand out in particular.

- The authors used a very elegant design where they let participants repeatedly learn and retrieve associations which enabled them to investigate changes across learning trials as well as to identify individual-specific learning events. Critically, this enabled them to investigate the relationship between representational changes associated with individual stimuli and behavioral learning of specific associations. I particularly liked the additional analysis looking at LR-3 to LR+3 (Fig2e).
- The authors showed in a convincing analysis that the initial representational overlap between stimuli was associated with the extent of remapping and that scene representations do differentiate between competitor and target objects.
- I really appreciate that the authors used individually generated regions-of-interest for hippocampal subregions which is critical and absolutely appropriate for the detailed analyses in this paper.

The study design is sound, the paper is very nicely written, and the figures are very clear and illustrate the findings well. In my opinion, the discussion and interpretation are balanced and fit the data well. Please find only one minor comment below.

Minor:

- The authors say that they identified the most anterior and most posterior slices of the hippocampal body based on the T2-weighted anatomical structure. Given that this defines which part of the hippocampus was analyzed it might be good to provide these landmarks for the reader.

Reviewer #3 (Remarks to the Author):

In the present manuscript, Guo and colleagues describe a reanalysis of an experiment described in prior publications which aimed to understand the role of the hippocampus in differentiating among related stimuli. The experiment involved 36 scene-object pairs, with half being highly similar to other pairs (e.g., two similar barn-guitar pairs). Learning was assessed across 6 rounds, with two conditions of interest being the 'learned round' (LR) during which the pair were reliably recalled with high confidence, and the 'inflection point' (IP) which describes the transition between the LR and the prior round. Pattern similarity analyses are reported in hippocampal subfields CA23DG and CA1, as well as parahippocampal place area (PPA) and early visual cortex (EVC). The authors report that CA23DG shows decorrelation of similar scene-object pairs (pairmates) at the IP relative to the pre-IP period (an effect also reported for PPA). This effect is reported to be 'abrupt' in that adjacent timepoints are more decorrelated than non-adjacent timepoints, and it is reported to be stronger for items that had initially more overlapping patterns, which the authors interpret as a representational 'repulsion' effect, in line with prior work from their group. Finally, the authors report evidence for increased pattern similarity between scenes and their associated target (non-competitor from pairmate) items with learning.

Overall, I think this is a well-designed and well-executed experiment. The results should be of reasonably broad interest, and particularly of interest to those focused on the mnemonic functions

of the hippocampus. I quite liked this paper. I do not have what I consider to be serious problems with or concerns about the core aspects of the experiment or manuscript, though I do have a number of issues and questions, which I enumerate below.

Major concerns:

1. In my opinion, the framing of this experiment in the Introduction does not match the nature and scope of the experiment particularly well, nor does it accurately portray the state of the field to which it contributes. The study is overwhelmingly framed in terms of studies in rodents pertaining to remapping of neuronal representations, and moreover, these studies primarily relate specifically to spatial representations and experimental paradigms. Though these mechanisms are no doubt relevant, there is a wealth of research in labs studying human hippocampal function and hippocampal signals indicating nonoverlap among experiences which are more directly relevant, and which are minimally discussed. I will not list all such labs and studies here, but to put it into perspective, the authors have a single sentence on lines 60-61 which very briefly pays credence to this work, compared to paragraphs devoted to discussions of remapping in rodents. I want to be clear that I am not pushing the authors to cite anyone's work in particular. Rather, the issue here is that the current framing comes across as presenting this study as among the only work that has been in a position to translate the ideas of hippocampal remapping in rodents to human participants, which is an inaccurate impression one could plausibly take away given the current framing. While I grant that the authors' design is quite elegant in being able to track hippocampal representations (and differentiation of hippocampal patterns) across numerous experiences, the basic premise of the work has been addressed to varying degrees by many prior studies in humans. In sum, I think that the paper would be stronger if the Introduction was reworked and reframed to more appropriately situate itself in the most relevant literature. To the authors' credit, I felt that the Discussion (while still fairly emphatic on rodent studies) was a bit more appropriately laid out in this regard.

2. I think that the quartile-based analysis could be more clearly motivated and described. I ultimately understood that the point of this is to demonstrate that the 4th quartile – the pairmates featuring the highest pattern similarity values – are the ones represented the most dissimilarly by CA23DG. However, this was not immediately clear, nor was it clear why it was done in this particular way. To this point, I am curious about why the data were split into quartiles instead of a median split, into thirds, or into any other dividing scheme. This analysis, in general, just feels somewhat arbitrarily done in the current presentation. Additionally, it is pretty striking that this differentiation effect is only present in the 4th quartile. That is, not only is there not a 'ramping up' of this divergence, there is not so much as a hint of it in the other 3 bins, even the 3rd quartile. Does this mean that the entire effect in this dataset is driven by only the most initially similar items? This warrants clarification of how many trials across the entire dataset this (evidently quite important) condition comprises, and additionally warrants some further unpacking of its implications.

3. I struggled with how to raise this point, as in my view it does not appear to be a result of anything the authors did or did not do. However, I find the scene-object similarity data (Figure 4) confusing. What makes sense to me is, as the authors describe, that the target item comes to be represented more similarly and the competitor item comes to be represented more dissimilarly with respect to the paired scene as a result of learning. That is, the directionality of the change is sensible. What I am struggling to understand is the difference between these values at the initial Pre-Learned stage. Unless I am misunderstanding this plot, competitor items are initially represented far more similarly to paired scenes than the target items to which they are actually paired. Per the example in Figure 4, that means that each barn evokes a pattern of activity in CA23DG that is considerably more similar to the wrong guitar than the guitar with which it is actually paired (and based on 'eyeballing' the figure, it seems that these pre-learning means likely differ from a correlation of zero). Simply put, I find this very difficult to understand, let alone interpret. And while this is not necessarily a crucial point in that target item patterns do increase and competitor item patterns do decrease, it is likely that these highly divergent Pre-Learning pattern similarity values have a strong influence on the reported interaction. That is, if they started near zero such as CA1, this interaction and indeed pre- versus post-learning differences

themselves might not be significant. While I do not think that there is any kind of re-analysis that would mitigate this concern (after all, the data are the data), I do think the authors could stand to unpack and make sense of this initial strong divergence in order to mitigate reader confusion with the interpretation (such as my own). What do we make of this learning-related change, in light of the fact that the correlations seemed to be so strongly in the other direction to begin with?

Minor concerns:

1. The authors use "CA23DG" and "CA3/dentate gyrus" in different parts of the paper. I suggest picking one of these terms and using it consistently.
2. Although clarity on this matter can be ascertained by reading the caption, I would suggest simply labeling the ROI (CA23DG) on Figure 2d. At first glance, one could reasonably assume this pertains to EVC given that it falls directly beneath the depiction of that ROI.
3. I honestly do not feel as if Figure 3c thematically fits with Figure 3a&b. The former two panels describe analyses as a function of lag, and lag is not at all incorporated into the latter panel. This sets up the expectation that the similarity quartiles are in fact temporally-relevant in some way. I would suggest moving panel c elsewhere (into a different figure, or into its own figure), or coming up with a way of clearly delineating between the lag analysis and the quartile analysis in a single figure.
4. Although I suspect that this is not terribly likely to have biased the data, did the authors look for potential systematic effects of scene or item memorability, or scene-item congruency in the data? That is, were some scenes or items particularly well remembered in this dataset? Is there any evidence that such an effect may have systematically influenced LR or the quartile distribution? Is there any evidence for memorability as a function of particular scene-item pairings? I do not mean to send the authors down an analytical 'rabbit hole' here, but it would be informative if these kinds of influences can be simply and easily ruled out.
5. In Figure 1e, I am unclear on why the data are being displayed as a cumulative number of pairs learned across the entire sample rather than averaged across the sample. It seems to me that the latter would be more informative.
6. This is very minor and a matter of personal preference, but I actually think that the readability of the paper could be somewhat improved by not abbreviating with 'LR' and 'IP'. On more than one occasion, I had to go back and remind myself what those acronyms stood for since they are very specific to this paradigm and not terms used by the field more broadly. Simply spelling out these terms – at least in the text, if not the figures – could help readers stay oriented, particularly given that they are not lengthy terms.

We thank the editor for the opportunity to revise our manuscript and the Reviewers for their excellent suggestions and questions. We have made revisions throughout the manuscript to address the Reviewers concerns and have conducted several new analyses, many included as new Supplementary Figures, that help clarify and reinforce our findings.

Reviewer #1

Guo and colleagues present novel fMRI results testing the notion that behavioral resolution of interference among similar memories is supported by hippocampal pattern repulsion or remapping. Specifically, patterns in the CA23DG subregion were found to remap as each participant successfully encoded scene-object pairs and maintained the correct representation for the rest of the experiment, without confusing it with competitor stimuli. This reconfiguration appears to be particularly strong for stimuli with high initial overlap. The data are novel and compelling, and the close link between behavioral markers of interference resolution and neural patterns is especially important and interesting. The manuscript is well-written and the statistical analyses are appropriate and described clearly. I do, however, have a few comments and questions which I hope the authors can address.

We thank the Reviewer for the positive feedback and insightful comments. As detailed below, the Reviewer's comments inspired several new analyses and revisions which have strengthened the manuscript.

Reviewer 1, Comment 1. The data in Figure 2C clearly suggest that pairmate similarity decreases from pre-IP to IP in CA23DG. However, I'm having a bit of difficulty understanding how these data relate to Fig 2E. If I'm interpreting panel E correctly, these data suggest that pre-inflection point (LR-1), the pairmate representations are dissimilar to the learned round representation. After the inflection point (LR+1 onwards), the pairmates then actually become more similar, and remain more similar. At the level of interpretation, would this suggest that the pairmate representations actually become more integrated post-IP when they are behaviorally more differentiated? Perhaps one way to address this question would be to compare pre-IP to post-IP representations. If the representations are repelled at the inflection point, one might imagine that the demonstrated sharp drop from pre-IP to IP should be sustained. This would be reflected in a similar magnitude of difference between pre-IP to post-IP as pre-IP to IP. Alternatively, if I misunderstood the analysis, more detail and description of the pattern of results in panel E would be welcome here to avoid confusion.

We apologize for the confusion in reconciling **Fig. 2c** and **2e**. To clarify: yes, **Fig. 2e** shows that when correlating the learned round (LR) with rounds that preceded the learned round (i.e., rounds LR-1, LR-2 and LR-3), pairmate similarity scores are relatively low (below 0), but when correlating the learned round with rounds that followed learning (LR+1, LR+2, LR+3), there is a relative increase in pairmate similarity scores. This asymmetry provides key evidence that representations at the learned round are specifically differentiated from competing memories in the pre-learned state (LR-1, LR-2, LR-3). Regarding the relationship between **Fig. 2e** and **2c**, one important point that was not sufficiently clear in the initial manuscript is that the LR-1 data point in **Fig. 2e** (i.e., similarity between LR-1 and LR) is the inflection point (i.e., the same data points shown in **2c**). We apologize that this was not sufficiently clear in the original manuscript. We have modified the figure and the caption (copied below) to make these points clearer. In panel **e** of the revised figure, we now denote the inflection point ("-1" on the x axis) with a filled circle (as in **2c**). All other circles are unfilled. This is also clarified in the caption. Note: in response to a suggestion from Reviewer 3, we now refer to the CA23DG ROI as "CA3/dentate gyrus" or as "CA3/DG" in the figures. The ROI itself is unchanged—just the label.

Figure 2. Pairmate similarity scores change at the behavioral inflection point. **a.** Regions of interest included CA3/dentate gyrus (CA3/DG) and CA1 in the hippocampus, the parahippocampal place area (PPA), and early visual cortex (EVC). **b.** Correlation matrix illustrating how pairmate similarity scores were computed at the behavioral inflection point. See Methods for details. **c.** Pairmate similarity scores at the behavioral inflection point (IP) and just prior to the inflection point (pre-IP) across different regions of interest (ROIs). Pairmate similarity scores significantly varied by ROI ($p = 0.009$) and there was a significant interaction between ROIs and behavioral state ($p = 0.011$). **d.** A permutation test (1,000 iterations) was performed by shuffling, within participants, the mapping between the behavioral inflection point and scene pairmates. In CA3/dentate gyrus the actual mean group-level pairmate similarity score at the IP was lower than 98.70% of the permuted mean similarity scores. **e.** Pairmate similarity scores calculated by correlating the learned round (LR) with each of the three preceding rounds ($-$ distance to LR) and each of the three succeeding rounds ($+$ distance to LR). [Note: the inflection point was defined as the correlation between the LR and the immediately preceding round (LR - 1); the inflection points are depicted by filled circles and are the same values as in c]. In CA3/dentate gyrus, pairmate similarity scores were significantly lower when the LR was correlated with preceding rounds compared to succeeding rounds ($p = 0.006$). The difference was not significant for any other ROIs (p 's ≥ 0.435). **f.** Conceptual illustration of a decrease in pairmate similarity scores from pre-IP to IP. In the pre-IP state (top panel), A_1 and A_2 are nearby in representational space. In the IP state (bottom panel), the representational distance between A_1 and A_2 has been exaggerated. When pairmates (e.g., A_1 and A_2) are farther apart in representational space than non-pairmates (e.g., A_1 and B_2) the pairmate similarity score will be *negative* (i.e., pairmate similarity < non-pairmate similarity), consistent with a repulsion of competing representations. Notes: * $p < .05$, ** $p < .01$, error bars reflect S.E.M.

Related to the Reviewer's second point, about whether pairmates become more similar after learning, this is an excellent question. As the Reviewer anticipated, after learning has occurred (LR, LR+1, LR+2, LR+3), representations in CA3/dentate gyrus tended to "stay" differentiated from the pre-learned state. The best way to appreciate this is by viewing the correlation matrix which expresses the representational similarity between all combinations of rounds, relative to the learned round (LR). We now include this correlation matrix as panel b in a new Supplemental Figure 2, copied below.

Supplementary Figure 2. Pairmate similarity scores as a function of timepoints and learning. **a.** Pairmate similarity scores at each timepoint for each region of interest (ROI). Each timepoint reflects correlations between successive scene exposure rounds [i.e., timepoint 1 = $r(\text{round 1, round 2})$, timepoint 2 = $r(\text{round 2, round 3})$, etc.]. CA1 is the only ROI that showed a significant main effect of timepoint (CA1: $F_{4,120} = 3.89$, $p = 0.005$, $\eta^2 = 0.09$; all other ROIs: p 's > 0.517). **b.** Pairmate similarity scores in CA3/dentate gyrus (CA3/DG) calculated by correlating all possible combinations of the scene exposure rounds, expressed in terms of distance relative to the learned round (LR) for each pairmate. Rounds that preceded the LR reflect rounds *before* learning occurred, whereas the LR and following rounds reflect rounds *after* learning occurred (i.e., high confidence correct performance on the associative memory test). Thus, the correlations can be grouped into 3 categories: correlations among 'before' rounds (-/-), correlations between 'before' and 'after' rounds (-/+), and correlations among 'after' rounds (+/+). **c.** CA3/dentate gyrus pairmate similarity scores averaged across all of the cells within each of the three categories (-/-, -/+, +/+). Pairmate similarity scores in the -/+ category were significantly below 0 ($t_{30} = -2.70$, $p = 0.011$, $d = 0.48$, CI = $[-0.004 \pm 0.003]$) and significantly lower than the -/- category ($t_{30} = 2.49$, $p = 0.018$, $d = 0.45$, CI = $[0.008 \pm 0.006]$). Notes: ** $p < .01$, * $p < .05$, ~ $p < .10$, error bars reflect S.E.M.

Related to the Reviewer's question, the key cells are those that reflect the correlations between the pre-learning rounds (LR-1, LR-2, LR-3) and all subsequent rounds (LR, LR+1, LR+2, LR+3). These cells are reflected in the rectangle at the bottom left of the figure (-3 to -1 on the x axis and 0 to 3 on the y axis). All of these cells reflect correlations between a pre-learned state and a learned state (which we label as "-/+"). Critically, the mean of the -/+ cells is significantly below 0, as shown in panel **c** ($t_{30} = -2.70$, $p = 0.011$, $d = 0.48$, CI = $[-0.004 \pm 0.003]$). This effect mirrors the inflection point analysis. In other words, if we correlate the LR with the immediately preceding round (i.e., the inflection point analysis; **Fig. 2c**), we get a very similar result compared to correlating *all post-learning rounds* with

all pre-learning rounds (the -/+ analysis in the new **Supplementary Figure 2c**). For correlations among the pre-learning rounds (-/- in the figure), the pairmate similarity scores are numerically positive ($t_{30} = 1.57$, $p = 0.127$, $d = 0.28$, $CI = [0.004 \pm 0.005]$) and significantly higher than -/+ correlations ($t_{30} = 2.49$, $p = 0.018$, $d = 0.45$, $CI = [0.008 \pm 0.006]$). This difference between -/- vs. -/+ is conceptually and quantitatively very similar to the difference between pre-IP and IP that we report in the main text (**Fig. 2c**).

Also relevant to the Reviewer's question are the correlations among representations in the post-learning state (i.e., correlations among LR, LR+1, LR+2, and LR+3). As can be seen in panels **b** and **c** of the **Supplementary Figure 2**, these +/+ correlations are somewhat higher than the -/+ correlations ($t_{30} = 1.98$, $p = 0.057$, $d = 0.36$, $CI = [0.004 \pm 0.005]$) but are only barely above 0 ($t_{30} = 0.45$, $p = 0.655$, $d = 0.08$, $CI = [0.0008 \pm 0.003]$). Interestingly, this means that there is relatively weak evidence for differentiation *among* the remapped representations (i.e., post learning); rather, remapped (post-learning) representations are differentiated *from the initial representations* (pre-learning) of competing memories. For this reason, we described the remapping result (in the original and revised manuscript) as follows (Line 210):

“...representations expressed at the learned round were systematically biased away from the initial representational position of competing memories.”

Regarding the Reviewer's question of whether a post-learning integration occurred, we are reluctant to go quite that far because the +/+ similarity in CA3/dentate gyrus still hovers around 0 and is numerically lower than similarity among the pre-learning representations (-/-). And, as described above, the post-learning representations remain differentiated from the pre-learning representations. However, as a final, related point, we do see a potentially interesting result in CA1, which is now included in the new **Supplementary Figure 2** (panel **a**). When comparing pairmate similarity scores across all five timepoints (without respect to behavior), we see that CA1 pairmate similarity scores significantly *increase* during the last couple of timepoints, which contrasts with CA3/dentate gyrus. Because this increase in CA1 pairmate similarity was not something we predicted in advance—and because it was not coupled to learning in the way that CA3/dentate gyrus changes were (see **Fig. 2c,e**)—we are reluctant to draw strong conclusions. That said, the pattern of data is consistent with the idea that CA1 may form ‘integrated’ representations (or at least some association between the scene pairmates) even after CA3/dentate gyrus remapping has occurred¹. Notably, our behavioral memory tests only probed subjects' ability to discriminate similar memories and therefore were not geared toward testing (behaviorally) for memory integration. It is certainly possible, but beyond the scope of the present study, that some brain regions (including subfields of the hippocampus) form differentiated representations while others simultaneously form integrated representations^{1,2}.

Reviewer 1, Comment 2. The link between the behavioral inflection point and neural pattern differentiation is key to interpreting these results. However, it would also be valuable to plot neural differentiation compared to pre-learning over time, similar to the behavioral plot in Fig 1c. Most participants reach the inflection point for most stimuli by the 3rd round of the experiment – do these representations continue drifting further from the start?

We believe the Reviewer is asking whether representations at LR, LR+1, LR+2, and LR+3 become progressively less similar to the pre-learning state (LR-3, LR-2, LR-1). This is an interesting question. We have included these data, from CA3/dentate gyrus, in a figure below. For this analysis, we correlated representations from all of the pre-learning rounds (LR-3, LR-2, LR-1) with post-learning representations expressed in terms of the distance to the learned round (LR). Thus, for the data at LR = 0, this reflects the mean of correlations between (LR,LR-1), (LR,LR-2), and (LR, LR-3). As can be seen, there is no clear, progressive decrease in similarity. The main effect of bin is not significant when including all bins or even when just including LR, LR+1, and LR+2, which are better powered (F 's ≥ 1.46 , p 's $\geq .241$). From our theoretical perspective, we also would not necessarily expect a progressive decrease in similarity. Rather, we predicted an abrupt change at the LR, as opposed to a gradual drift away from the pre-learning state. Given that these data are a bit noisy/underpowered and do not afford strong inference, we have not included them as a separate figure in the supplement. However, the means shown in the figure below are simply a weighted average of the values from the correlation matrix shown in the new **Supplementary Figure 1b** (see above). Thus, these data can also be roughly inferred from that figure.

Reviewer 1, Comment 3. Related to the point above, the rank order correlation analysis is interesting and supports the interpretation that representational structure is negatively related across successive timepoints. However, it would be informative to see the successive rank N+1 correlation values plotted across time, rather than averaged per lag. Fig 3 shows lag 1 vs. lag 2, but not the change in similarity over time. It would be helpful to see lag 1 and lag 2 values split by round (where possible).

Below we have split the Lag 1 and Lag 2 data by round. Specifically, each plot shows the rank order correlation for Lag 1 (left) or Lag 2 (right) as a function of ROI (CA1 vs. CA3/dentate gyrus) and the specific timepoints on which the correlations are based. For the Lag 1 data, there is a main effect of ROI (lower rank correlations for CA3/dentate gyrus than CA1: $F_{1,30} = 13.90$, $p < 0.001$, $\eta^2 = 0.06$), which is just a re-expression of the result we report in the main text, but there is no effect of timepoint ($F_{2,60} = 0.86$, $p = 0.429$, $\eta^2 = 0.01$) nor, of most relevance, an interaction between ROI and timepoint ($F_{2,60} = 1.85$, $p = 0.166$, $\eta^2 = 0.02$). In other words, while the data are slightly noisy, the negative rank order correlation that we observed did not significantly vary by timepoint. For Lag 2, there were no main effects or interactions (p 's > 0.441). We have not added these data to the Supplement because it is not clear that these data afford much inference, but we are happy to do so if the Reviewer believes they would be useful. It is also possible we have misunderstood the Reviewer's question as the Reviewer mentions the "change in similarity over time," which is not quite what the figure below shows (the below would be better described as *differences* in the change in similarity over time). More directly visualizing the change in pattern similarity over time in a way that aggregates over items and subjects is not straightforward, unfortunately. That said, the purpose of **Fig 3c** is to help provide a visualization of this relationship and to connect it with the behavioral inflection point.

Reviewer 1, Comment 4. The finding that pairmates with the highest pre-IP similarity became more differentiated is very interesting. This analysis is thoughtful and gets at the issue of what's driving the observed remapping. However, it would also be helpful to see a plot of the distribution of all the pre-IP pairmate similarity values (i.e., how much did the pre-IP values for each quartile differ across participants relative to their IP representations). Further, were the quartiles calculated within each participant and each subregion? The calculation approach should be included in the Methods.

We have added a new **Supplementary Figure 4** which shows the mean pairmate similarity scores for CA3/dentate gyrus and CA1 for each pre-IP quartile alongside the corresponding scores at the IP. To provide a sense of the distribution of data, we include dots for individual subjects. However, as we emphasize in the figure caption, it is important to keep in mind that direct comparisons between the pre-IP and IP values are not valid (given that the pre-IP scores were sorted by value but the IP scores were not). Rather, this figure is intended to visualize the distribution

of values within each quartile, separately for pre-IP and IP, as opposed to a visualization of the amount of change (which is why we have not included lines to connect pre-IP and IP values that correspond to the same subject).

Supplementary Figure 4. Distributions of pairmate similarity scores at the pre-inflection point (pre-IP) and inflection point (IP), as a function of pre-IP similarity (related to Fig. 3c). Mean pairmate similarity scores at the pre-IP were binned into quartiles, separately for each subject and for CA3/dentate gyrus (CA3/DG) and CA1. Pre-IP data (blue dots) show the distribution (across subjects) of the pairmate similarity scores at each pre-IP bin. IP data (orange dots) show the distribution (across subjects) of pairmate similarity scores at the inflection point as a function of the pre-IP pairmate similarity level (1st quartile = lowest pre-IP similarity, 4th quartile = highest pre-IP similarity). Note: direct comparison of pre-IP versus IP values at each bin is not statistically valid given that the pre-IP data, but not the IP data, were binned by value (quartiles).

We have also revised the methods (line 626) to clarify that the pre-IP similarity bins were computed within subject and separately for each ROI. The revised text is copied below:

“...The quartile analysis was performed within subject and separately for CA3/dentate gyrus and CA1.”

Reviewer 1, Comment 5. The results in Fig 4 are compelling and suggest that learning reconfigured object-scene associations as well. However, it's not immediately clear to me why the object exposure scans were averaged across the first and final rounds. Since the two object rounds were in the first and final blocks, they could also be classified as pre- and post-learning. The split of object representations into early and late scans seems even more important given the interaction with behavioral state. Further, do object representations also show any reconfiguration from pre- to post-learning, akin to scenes? If both first- and final-round object representations show the same pattern, this does not necessarily change the interpretation. However, it could suggest something interesting about the nature of these associations, such that scenes can reconfigure object representations, but not vice versa.

The Reviewer raises another great question. We did look at this point during our initial analyses but opted against discussing it in the original manuscript because we did not observe significant effects of object block (first vs. final) and we were concerned about adding too much complexity. That said, the data are qualitatively very much consistent with what the Reviewer alludes to. Specifically, the critical interaction we report in CA3/dentate gyrus between object relevance (target, competitor) and behavioral state (pre-learned, learned) is highly significant when considering object data from the final block ($F_{1,30} = 12.65, p = 0.001, \eta^2 = 0.07$) but is not significant when considering object data from the first block ($F_{1,30} = 0.48, p = 0.495, \eta^2 = 0.003$). The interaction, however, between object relevance, behavioral state, and object block (first, final) did not quite reach significance in CA3/dentate gyrus ($F_{1,30} = 3.14, p = 0.086, \eta^2 = 0.01$). Still, we believe these are interesting data and we have added a new **Supplementary Figure 5** which shows scene-object similarity as a function of object relevance and behavioral state, separately for data from each object block (first, last) and for CA3/dentate gyrus and CA1 (see below). Notably, the interaction was not significant in CA1 for either block (p 's > 0.059). Finally, we do not report whether object-object correlations (e.g., between competing objects) change from the first to last block because all of our pattern similarity analyses are based on correlations *across scans* and there was only one object scan at the beginning of the fMRI session and only one at the end of the fMRI session.

Supplementary Figure 5. Scene-object similarity as a function of object relevance (target, competitor), ROI (CA3/dentate gyrus, CA1), behavioral state (pre-learned, learned), and object run (first, final). The 3-way interaction between object relevance, behavioral state, and object run was not significant for CA3/dentate gyrus (CA3/DG: $F_{1,30} = 3.14$, $p = 0.086$, $\eta^2 = 0.01$) or for CA1 ($F_{1,30} = 3.90$, $p = 0.057$, $\eta^2 = 0.01$). When considering each object run separately, CA3/dentate gyrus exhibited a significant interaction between object relevance and behavioral state for data from the final object run ($F_{1,30} = 12.65$, $p = 0.001$, $\eta^2 = 0.07$), but not from the first object run ($F_{1,30} = 0.48$, $p = 0.495$, $\eta^2 = 0.003$). CA1 did not exhibit a significant interaction between object relevance and behavioral state for data from either object run (p 's > 0.059). Note: error bars reflect S.E.M.

Reviewer 1, Comment 6. On p. 13, the authors state that the dissociation in representations across brain regions suggests that CA23DG did not simply inherit the representational structure from EVC. The data support this view, but I found this to be an interesting issue that could be tested even more directly - specifically, is higher overlap between memories in EVC in any way related to repulsion (or integration) in CA23DG?

We agree and appreciate the suggestion. We replicated the rank order correlation (Lag 1) but this time correlated representational structure in EVC at timepoint N with representational structure in CA3/dentate gyrus at timepoint N+1. The correlation was not significant ($t_{30} = -0.12$, $p = 0.902$, $d = 0.02$, $CI = [-0.003 \pm 0.05]$). In other words, representational structure in CA3/dentate gyrus at a given timepoint was predicted by the *preceding* representational structure in CA3/dentate gyrus, but not in EVC. We have added this result to the main text (Line 250), as follows:

“Additionally, rank order correlations did not differ from 0 when representational structure at timepoint N was defined from EVC and representational structure at timepoint N+1 (lag 1) or N+2 (lag 2) was defined from CA3/dentate gyrus (lag 1: \$t_{30} = -0.12\$, \$p = 0.902\$, \$d = 0.02\$, \$CI = [-0.003 \pm 0.05]\$; lag 2: \$t_{30} = -0.22\$, \$p = 0.825\$, \$d = 0.04\$, \$CI = [-0.005 \pm 0.05]\$.”

We also replicated the analysis from **Fig. 3c**, but this time defined the pre-IP quartiles *using EVC data* and then computed the pairwise similarity scores at the IP for CA3/dentate gyrus, CA1 and EVC. These data are shown below. When binned by pre-IP similarity in EVC, there was no longer a significant difference between CA1 and CA23DG in the highest pre-IP bin ($t_{30} = 0.39$, $p = 0.698$, $d = 0.07$, $CI = [0.006 \pm 0.03]$). Various statistical analyses all failed to reveal any significant effects when binning by pre-IP scores in EVC (either for CA3/dentate gyrus, CA1, or for EVC itself). We have not added the figure below to the Supplement, but are happy to do so if the Reviewer feels it would be helpful.

Reviewer #2

Guo and colleagues present timely and solid work on remapping in human hippocampal subfields CA3 and dentate gyrus and its role in the resolution of memory interference.

In my opinion, the following strengths stand out in particular.

- The authors used a very elegant design where they let participants repeatedly learn and retrieve associations which enabled them to investigate changes across learning trials as well as to identify individual-specific learning events. Critically, this enabled them to investigate the relationship between representational changes associated with individual stimuli and behavioral learning of specific associations. I particularly liked the additional analysis looking at LR-3 to LR+3 (Fig2e).

- The authors showed in a convincing analysis that the initial representational overlap between stimuli was associated with the extent of remapping and that scene representations do differentiate between competitor and target objects.

- I really appreciate that the authors used individually generated regions-of-interest for hippocampal subregions which is critical and absolutely appropriate for the detailed analyses in this paper.

The study design is sound, the paper is very nicely written, and the figures are very clear and illustrate the findings well. In my opinion, the discussion and interpretation are balanced and fit the data well. Please find only one minor comment below.

We thank the Reviewer for this positive feedback!

Reviewer 2, Comment 1. The authors say that they identified the most anterior and most posterior slices of the hippocampal body based on the T2-weighted anatomical structure. Given that this defines which part of the hippocampus was analyzed it might be good to provide these landmarks for the reader.

We agree. We have added a new **Supplementary Figure 6** which shows the critical landmarks for segmenting the hippocampal body (the anterior and posterior boundaries) on a high-resolution anatomical scan from a representative subject. The new figure is copied below.

Supplementary Figure 6. Illustration of anterior and posterior boundaries for the hippocampal body from a sample participant.

Reviewer #3

In the present manuscript, Guo and colleagues describe a reanalysis of an experiment described in prior publications which aimed to understand the role of the hippocampus in differentiating among related stimuli. The experiment involved 36 scene-object pairs, with half being highly similar to other pairs (e.g., two similar barn-guitar pairs). Learning was assessed across 6 rounds, with two conditions of interest being the 'learned round' (LR) during which the pair were reliably recalled with high confidence, and the 'inflection point' (IP) which describes the transition between the LR and the prior round. Pattern similarity analyses are reported in hippocampal subfields CA23DG and CA1, as well as parahippocampal place area (PPA) and early visual cortex (EVC). The authors report that CA23DG shows decorrelation of similar scene-object pairs (pairmates) at the IP relative to the pre-IP period (an effect also reported for PPA). This effect is reported to be 'abrupt' in that adjacent timepoints are more decorrelated than non-adjacent timepoints, and it is reported to be stronger for items that had initially more overlapping patterns, which the authors interpret as a representational 'repulsion' effect, in line with prior work from their group. Finally, the authors report evidence for increased pattern similarity between scenes and their associated target (non-competitor from pairmate) items with learning.

Overall, I think this is a well-designed and well-executed experiment. The results should be of reasonably broad interest, and particularly of interest to those focused on the mnemonic functions of the hippocampus. I quite liked this paper. I do not have what I consider to be serious problems with or concerns about the core aspects of the experiment or manuscript, though I do have a number of issues and questions, which I enumerate below.

We thank the Reviewer for the positive feedback and insightful comments. As detailed below, the Reviewer's comments led to several revisions and new analyses/figures which have strengthened the manuscript. As a point of clarification, however, the current manuscript is not a re-analysis of a previously-described experiment. This is a new experiment, but it was closely modeled after a prior study (Favila, Chanales, & Kuhl, 2016).

Major concerns:

Reviewer 3, Comment 1. In my opinion, the framing of this experiment in the Introduction does not match the nature and scope of the experiment particularly well, nor does it accurately portray the state of the field to which it contributes. The study is overwhelmingly framed in terms of studies in rodents pertaining to remapping of neuronal representations, and moreover, these studies primarily relate specifically to spatial representations and experimental paradigms. Though these mechanisms are no doubt relevant, there is a wealth of research in labs studying human hippocampal function and hippocampal signals indicating nonoverlap among experiences which are more directly relevant, and which are minimally discussed. I will not list all such labs and studies here, but to put it into perspective, the authors have a single sentence on lines 60-61 which very briefly pays credence to this work, compared to paragraphs devoted to discussions of remapping in rodents. I want to be clear that I am not pushing the authors to cite anyone's work in particular. Rather, the issue here is that the current framing comes across as presenting this study as among the only work that has been in a position to translate the ideas of hippocampal remapping in rodents to human participants, which is an inaccurate impression one could plausibly take away given the current framing. While I grant that the authors' design is quite elegant in being able to track hippocampal representations (and differentiation of hippocampal patterns) across numerous experiences, the basic premise of the work has been addressed to varying degrees by many prior studies in humans. In sum, I think that the paper would be stronger if the Introduction was reworked and reframed to more appropriately situate itself in the most relevant literature. To the authors' credit, I felt that the Discussion (while still fairly emphatic on rodent studies) was a bit more appropriately laid out in this regard.

We appreciate the Reviewer's feedback and perspective on the Introduction. We agree that the presentation of relevant human fMRI studies was fairly minimal and could give a mis-impression about the state of the field. Our rationale for framing the paper in terms of rodent remapping studies was (and remains) that these studies are the primary *inspiration* for the current study. For example, Colgin, Moser and Moser (2008) very clearly argue that remapping, as observed in place cells in the rodent hippocampus, may be critical to resolving episodic memory interference. In fact, one of the figures from that paper presents a pair of sample images (extremely similar scenes) that are essentially identical to the kinds of paired scene stimuli we used. Thus, we felt compelled to acknowledge this source of inspiration. Moreover, we also think there is tremendous value in bridging the rodent and human literatures. Remapping, after all, is a phenomenon that was discovered in rodents and there remain relatively few human fMRI studies that have explicitly framed their findings as evidence of remapping. Thus, we have opted against a major reframing of the Introduction, but we have modified the 2nd and 3rd paragraphs of the Introduction to more

clearly highlight the state of human fMRI studies in relation to evidence from rodent studies and to more clearly articulate how the current study builds on prior human fMRI studies. We believe these changes (a) address the Reviewer's concern, (b) have improved the Introduction, and (c) remain true to the fact that we took direct inspiration from studies in rodents. For convenience, we have copied the 2nd and 3rd paragraphs of the Introduction below, with the new/revised text in blue.

“One of the most important properties of remapping in the rodent hippocampus is that it is characterized by abrupt transitions between representations⁹⁻¹². These abrupt transitions, evidenced by decorrelations in patterns of neural activity, have most typically been observed as a function of the degree of environmental change^{9,11}. However, abrupt remapping can also occur as a function of experience with a new environment^{10,12}. Evidence of experience-dependent remapping^{6,13,14} suggests an important point: that remapping fundamentally reflects changes in internal representations, as opposed to changes in environmental states^{15,16}. An emphasis on internal representations lends itself well to human episodic memory in that it suggests that hippocampal remapping should occur as memories change. More specifically, this perspective makes the critical prediction that when two events are highly similar, hippocampal remapping will occur if, and when, corresponding memories become distinct. To date, a number of human fMRI studies have observed experience-dependent decorrelations in hippocampal representations of similar memories¹⁷⁻²² and/or have linked hippocampal pattern overlap to memory interference^{20,23-25}. However, to test the prediction that hippocampal activity patterns abruptly remap when memory interference is resolved it is necessary to precisely track changes in memories as a function of temporally-specific changes in hippocampal representations. Critically, standard approaches of averaging fMRI data across different stimuli (memories), stimulus repetitions, and/or participants can easily obscure or wash out abrupt changes in hippocampal representations if the timing of those changes varies across memories or participants.”

Evidence of place cell remapping in rodents also motivates specific predictions regarding the relative contributions of hippocampal subfields, with a major distinction being between CA3/dentate gyrus and CA1^{8,26,27}. In general, CA3 and dentate gyrus are thought to be more important than CA1 for discriminating between similar stimuli^{16,28,29,27,30,31} and remapping has been shown to occur more abruptly in CA3 than in CA1^{10,12,32}. High-resolution fMRI studies in humans have also tested for and confirmed distinctions between these subfields. For example, fMRI studies have found that, relative to CA1, activity patterns/responses in CA3 and dentate gyrus are more sensitive to subtle differences between similar memories^{17,19,33,34} or spatial environments^{23,24,33}. Moreover, responses in human CA3/dentate gyrus have specifically been linked to behavioral discrimination of similar memories^{23,24,35}. However, these studies have not directly established a link between temporally abrupt remapping in CA3/dentate gyrus and changes in corresponding episodic memories.”

Reviewer 3, Comment 2. I think that the quartile-based analysis could be more clearly motivated and described. I ultimately understood that the point of this is to demonstrate that the 4th quartile – the pairmates featuring the highest pattern similarity values – are the ones represented the most dissimilarly by CA23DG. However, this was not immediately clear, nor was it clear why it was done in this particular way. To this point, I am curious about why the data were split into quartiles instead of a median split, into thirds, or into any other dividing scheme. This analysis, in general, just feels somewhat arbitrarily done in the current presentation. Additionally, it is pretty striking that this differentiation effect is only present in the 4th quartile. That is, not only is there not a ‘ramping up’ of this divergence, there is not so much as a hint of it in the other 3 bins, even the 3rd quartile. Does this mean that the entire effect in this dataset is driven by only the most initially similar items? This warrants clarification of how many trials across the entire dataset this (evidently quite important) condition comprises, and additionally warrants some further unpacking of its implications.

The Reviewer raises two important questions: (1) why divide into quartiles? and (2) was the repulsion effect entirely driven by the 4th quartile?

Related to the first question (why divide into quartiles?): the purpose of **Fig. 3c** (the quartile analysis) was to provide a visualization of the relationship between representational structure across time and to connect this relationship to the critical behavioral measure of learning. We felt that a median split (two bins) would be rather coarse. In particular,

although detailed consideration of this point is beyond the scope of the present study, we were mindful of prior evidence that has identified non-monotonic relationships between memory activation/competition and plasticity³. Obviously, potential non-monotonic relationships would be impossible to observe with only two bins. Using a quartile analysis (as was done in a conceptually-related study by Newman et al.⁴) provides a better opportunity to capture potentially non-monotonic relationships. To help address the Reviewer's concern about the number of bins, we have added a new **Supplementary Figure 3**, copied below, which shows two alternative binning schemes (terciles and quintiles). Qualitatively, the pattern is very similar to the quartile binning: CA3/dentate gyrus pairmate similarity scores are numerically lowest for the highest pre-IP bin (*t*-test vs. 0 for highest tercile: $t_{30} = -1.83$, $p = .077$, $d = 0.33$, $CI = [-0.012 \pm 0.014]$; *t*-test vs. 0 for highest quintile: $t_{30} = -2.30$, $p = 0.28$, $d = 0.41$, $CI = [-0.023 \pm 0.020]$). Additionally, the difference between CA3/dentate gyrus and CA1 is significant (and only significant) for the highest pre-IP bin (*t*-test for highest tercile: $t_{30} = -2.22$, $p = .034$, $d = 0.40$, $CI = [-0.020 \pm 0.018]$; *t*-test for highest quintile: $t_{30} = -2.18$, $p = .037$, $d = 0.39$, $CI = [-0.027 \pm 0.025]$). Thus, the key result is quite consistent across different binning procedures. Again, while detailed consideration of the shape of the function is beyond the scope of the present study, we felt it would be useful to provide a visualization that would at least allow for qualitative inspection of the function. For example, whether we bin data by terciles, quartiles, or quintiles, we do see hints of a non-monotonic relationship between pre-IP similarity and IP similarity in CA3/dentate gyrus. Given that we plan to make all data publicly available, this could potentially be explored in more targeted analyses by any research groups that are specifically interested in potential non-monotonic relationships.

Supplementary Figure 3. Pairmate similarity scores at the inflection point (IP) as a function of relative pairmate similarity scores at the pre-inflection point (pre-IP) (related to Fig. 3c). Left: pre-IP pairmate similarity scores binned into terciles (from lowest to highest). Right: pre-IP pairmate similarity scores binned into quintiles (from lowest to highest). For both analyses, binning was performed within-subject and separately for CA3/dentate gyrus (CA3/DG) and CA1. When binned by terciles or quintiles, pairmate similarity scores in CA3/dentate gyrus were significantly lower than in CA1 at the highest pre-IP bin (terciles: $t_{30} = -2.22$, $p = .034$, $d = 0.40$, $CI = [-0.020 \pm 0.018]$; quintiles: $t_{30} = -2.18$, $p = .037$, $d = 0.39$, $CI = [-0.027 \pm 0.025]$). When binned by terciles, pairmate similarity scores in CA3/dentate gyrus were marginally below 0 for the highest pre-IP bin (*t*-test vs. 0: $t_{30} = -1.83$, $p = .077$, $d = 0.33$, $CI = [-0.012 \pm 0.014]$). When binned by quintiles, pairmate similarity scores in CA3/dentate gyrus were significantly below 0 for the highest pre-IP bin (*t*-test vs. 0: $t_{30} = -2.30$, $p = 0.28$, $d = 0.41$, $CI = [-0.023 \pm 0.020]$). Notes: * $p < .05$, error bars reflect S.E.M.

We have added the following new text to the Methods section to better justify the rationale for binning by quartiles (line 628):

“The decision to divide pre-IP pairmate similarity scores into four bins was motivated by evidence, from conceptually-related studies, of non-monotonic relationships between initial memory activation/competition and experience-dependent plasticity^{17,18}. While formally testing for non-monotonic relationships was beyond the scope of the current study, the goal was to allow for qualitative inspection of the relationship. Notably, similar results were obtained when pre-IP pairmate similarity scores were binned by terciles or quintiles (Supplementary Figure 3).”

In the Results section (line 257), we also now orient the reader to the Methods and the new Supplementary Figure as shown below (revised text in blue):

“...we binned all pairmates, by quartiles, according to pre-IP pairmate similarity scores, with the 4th quartile representing pairmates with the highest pre-IP pairmate similarity scores (see Methods for additional rationale; see Supplementary Figure 3 for alternative binning procedures).”

Related to the Reviewer’s second question (was the repulsion effect entirely driven by the 4th quartile?), it is true, as we describe above, that the repulsion effect (the negative pairmate similarity score at the inflection point) was numerically strongest for the highest pre-IP pairmate similarity bin—a result which we believe is quite striking and theoretically important. Again, this was true whether pre-IP similarity values were divided into terciles, quartiles, or quintiles. In fact, it was also true if the data were median split (2 bins) or divided into 6 bins. That said—and as we note above—we also saw hints of non-monotonic relationships, with some numerically below 0 effects at lower pre-IP levels. For the quartile analysis, pairmate similarity scores at the IP were numerically below 0 for quartiles 1 and 2, in addition to quartile 4. Likewise, as can be seen in the new **Supplementary Figure 3**, pairmate similarity scores at the IP were numerically below 0 for the 1st tercile (in addition to the 3rd) and for the 1st and 2nd quintiles (in addition to the 5th). Notably, for each of these different binning schemes, *most* of the bins were numerically below 0. Thus, it is not quite accurate to say that the “entire effect” was driven by only the highest bin. Rather, the effect was numerically strongest for the most similar items, but other bins certainly contributed to the overall negative pairmate similarity score. Ultimately, there are some complexities to this data (potential non-monotonic effects) that are intriguing and of potential theoretical significance, but we believe they are beyond the scope (and statistical power) of the current study. Nonetheless, we believe the inclusion of the new Supplementary Figure is useful and should help address the Reviewer’s concern.

Finally, we also now report in the Methods section the mean number of scene pairmates in each quartile (line 627):

“The mean number of pairmates included in each pre-IP bin were 3.42, 2.90, 3.10, and 2.55 for quartiles 1-4, respectively.”

Reviewer 3, Comment 3. I struggled with how to raise this point, as in my view it does not appear to be a result of anything the authors did or did not do. However, I find the scene-object similarity data (Figure 4) confusing. What makes sense to me is, as the authors describe, that the target item comes to be represented more similarly and the competitor item comes to be represented more dissimilarly with respect to the paired scene as a result of learning. That is, the directionality of the change is sensible. What I am struggling to understand is the difference between these values at the initial Pre-Learned stage. Unless I am misunderstanding this plot, competitor items are initially represented far more similarly to paired scenes than the target items to which they are actually paired. Per the example in Figure 4, that means that each barn evokes a pattern of activity in CA23DG that is considerably more similar to the wrong guitar than the guitar with which it is actually paired (and based on ‘eyeballing’ the figure, it seems that these pre-learning means likely differ from a correlation of zero). Simply put, I find this very difficult to understand, let alone interpret. And while this is not necessarily a crucial point in that target item patterns do increase and competitor item patterns do decrease, it is likely that these highly divergent Pre-Learning pattern similarity values have a strong influence on the reported interaction. That is, if they started near zero such as CA1, this interaction and indeed pre- versus post-learning differences themselves might not be significant. While I do not think that there is any kind of re-analysis that would mitigate this concern (after all, the data are the data), I do think the authors could stand to unpack and make sense of this initial strong divergence in order to mitigate reader confusion with the interpretation (such as my own). What do we make of this learning-related change, in light of the fact that the correlations seemed to be so strongly in the other direction to begin with?

We completely agree that the data point to which the Reviewer refers is surprising. Specifically, in the pre-learned state, CA3/dentate gyrus scene representations exhibit stronger similarity to the competitor object than to the target object. This difference is statistically significant ($t_{30} = 2.70$, $p = 0.011$, $d = 0.48$, $CI = [0.012 \pm 0.009]$). We definitely took note of this result before submitting the original manuscript and considered potential explanations. Because these were post-hoc explanations—and because of recommended word limits for the manuscript—we did not comment on or attempt to explain this result in our original submission. However, we welcome the opportunity to do so here. We think it may be informative to consider these scene-object data in relation to our other findings that (a) repulsion of CA3/dentate gyrus activity patterns is positively related to learning, and (b) overlap among CA3/dentate gyrus activity patterns triggers repulsion. Putting these two results together, it suggests a potentially counterintuitive relationship wherein CA3/dentate gyrus pattern overlap leads to repulsion, which *in turn* is associated with learning. By this account, when scene representations get ‘too close’ to competing object associations (the competitor object), this triggers a correction in favor of the target object association which is evidenced in the subsequent round (the learned round). By analogy, this is similar to a driver whose car drifts toward the lane of oncoming traffic and then

corrects for this drift by swerving in the opposite direction. Obviously, this is a post-hoc account and we opted not to push this account in the original version of the manuscript, but it is an interesting account and it makes direct contact with recent empirical findings and theoretical frameworks. We have made the following revisions:

We now specifically highlight this finding in the Results section (line 308):

“Interestingly, for CA3/dentate gyrus, scene representations in the pre-learned state were significantly more similar to competitor objects than to target objects ($t_{30} = 2.70$, $p = 0.011$, $d = 0.48$, $CI = [0.012 \pm 0.009]$).”

And we now offer the following speculative, but theoretically-grounded account in the Discussion (line 374; revised text in blue):

“While our study does not afford inferences about the causal relationship between repulsion and learning, the idea that repulsion (or remapping more generally) is triggered by representational overlap, combined with the fact that remapping was associated with learning, is consistent with the possibility that repulsion of CA3/dentate gyrus representations is a causal factor in learning. This account also offers a potential explanation for an otherwise surprising finding: that CA3/dentate gyrus scene representations from the round that immediately preceded learning (LR-1) were significantly more similar to competitor objects than to target objects. Although speculative, it is possible that when a given scene activated the ‘wrong’ object association (at LR-1), this actively triggered a correction in favor of the target object association that supported learning. This account is consistent with evidence that prediction errors can powerfully drive episodic memory^{45,46} as well as differentiation of hippocampal activity patterns¹⁹. More broadly—and consistent with our findings, in general—prediction errors may induce abrupt state changes in the hippocampus that facilitate the separation of episodic memories⁴⁷.”

Minor concerns:

Reviewer 3, Comment 4. The authors use “CA23DG” and “CA3/dentate gyrus” in different parts of the paper. I suggest picking one of these terms and using it consistently.

We apologize for the confusion. Although we did not make this point clear in the original manuscript, we used CA23DG to refer to the specific ROI used in the current study, but we used CA3/dentate gyrus to collectively refer to these specific structures (independent of how those structures were identified in any given study). However, we agree this causes some confusion. We have therefore followed the Reviewer’s advice and have changed to using CA3/dentate gyrus throughout the manuscript. We now note in the Methods section that the CA3/dentate gyrus ROI also includes CA2.

Reviewer 3, Comment 4. Although clarity on this matter can be ascertained by reading the caption, I would suggest simply labeling the ROI (CA23DG) on Figure 2d. At first glance, one could reasonably assume this pertains to EVC given that it falls directly beneath the depiction of that ROI.

Agreed. We have added the text.

Reviewer 3, Comment 5. I honestly do not feel as if Figure 3c thematically fits with Figure 3a&b. The former two panels describe analyses as a function of lag, and lag is not at all incorporated into the latter panel. This sets up the expectation that the similarity quartiles are in fact temporally-relevant in some way. I would suggest moving panel c elsewhere (into a different figure, or into its own figure), or coming up with a way of clearly delineating between the lag analysis and the quartile analysis in a single figure.

We apologize that the link between these panels was not sufficiently clear. The common element in both analyses is that they consider pairwise similarity scores at a given point in time as a function of pairwise similarity scores at a preceding point in time. For 3a&b, the critical data are the lag 1 data (i.e., the relationship across successive timepoints), with the lag 2 data functioning as an important control to confirm that the negative rank order correlation was selective to successive timepoints. Figure 3c is intended to help visualize this relationship in representational structure across successive timepoints (lag 1) and to specifically connect this to the change from the pre-IP to the inflection point. This provides a visualization of the relationship between representational structure at the pre-IP and

at the inflection point. Thus, we view these figures as conceptually related and would therefore prefer to retain the current organization within a single figure. However, we have clarified how/why these analyses are related in the text.

We have modified the Results section (line 239), as follows (revised text in blue):

“Strikingly, the rank correlation in CA3/dentate gyrus was significantly negative ($t_{30} = -2.99$, $p = 0.006$, $d = 0.54$, $CI = [-0.06 \pm 0.04]$). In contrast, the rank correlation in CA1 was significantly positive ($t_{30} = 2.11$, $p = 0.043$, $d = 0.38$, $CI = [0.06 \pm 0.05]$). The difference between CA3/dentate gyrus and CA1 was also significant ($t_{30} = 3.73$, $p < 0.001$, $d = 0.67$, $CI = [0.12 \pm 0.06]$). Importantly, the negative correlation in CA3/dentate gyrus cannot be explained by regression to the mean (see Methods). As a control, we also tested correlations at a lag of 2 [$r(\text{timepoint } N, \text{timepoint } N+2)$]; however lag 2 correlations did not significantly differ from 0 for either CA3/dentate gyrus ($t_{30} = -0.71$, $p = 0.485$, $d = 0.13$, $CI = [-0.02 \pm 0.05]$) or CA1 ($t_{30} = -1.60$, $p = 0.120$, $d = 0.29$, $CI = [-0.04 \pm 0.05]$). The interaction between lag (1, 2) and ROI (CA3/dentate gyrus, CA1) was also significant ($F_{1,30} = 7.09$, $p = 0.012$, $\eta^2 = 0.06$). Thus, for CA3/dentate gyrus and CA1, representational structure at a given time point specifically predicted representational structure at a successive timepoint. Rank correlations did not differ from 0 in either PPA or EVC, either for lag 1 or lag 2 ($|t_{30}| \leq 1.12$, p 's ≥ 0.272 , d 's ≤ 0.20). Additionally, rank order correlations did not differ from 0 when representational structure at timepoint N was defined from EVC and representational structure at timepoint N+1 (lag 1) or N+2 (lag 2) was defined from CA3/dentate gyrus (lag 1: \$t_{30} = -0.12\$, \$p = 0.902\$, \$d = 0.02\$, \$CI = [-0.003 \pm 0.05]\$; lag 2: \$t_{30} = -0.22\$, \$p = 0.825\$, \$d = 0.04\$, \$CI = [-0.005 \pm 0.05]\$ ).”

To better visualize the relationship in representational structure across successive timepoints—and to specifically connect this relationship to learning (as in Fig. 2c)—we computed pairwise similarity scores at the inflection point as a function of pre-IP pairwise similarity scores. Specifically, we binned all pairmates, by quartiles, according to pre-IP pairwise similarity scores, with the 4th quartile representing pairmates with the highest pre-IP pairwise similarity scores (see Methods for additional rationale: see \Supplementary Figure 3 for alternative binning procedures). We then computed the mean pairwise similarity scores at the inflection point for each of the pre-IP quartiles. Again, this analysis was separately performed for CA3/dentate gyrus and CA1. An ANOVA with factors of ROI (CA3/dentate gyrus, CA1) and pairwise similarity scores at the pre-IP (4 quartiles) revealed a significant interaction (\$F_{3,90} = 3.19\$, \$p = 0.027\$, \$\eta^2 = 0.03\$ ), indicating that pre-IP representational overlap was differentially related to representational overlap at the inflection point for CA3/dentate gyrus versus CA1. Critically, this interaction was driven by a marked difference between CA3/dentate gyrus and CA1 when considering the bin with the highest overlap at the pre-IP (i.e., 4th quartile: \$t_{30} = -2.87\$, \$p = 0.008\$, \$d = 0.51\$, \$CI = [-0.03 \pm 0.02]\$, Fig. 3c). For CA3/dentate gyrus, pairwise similarity scores at the inflection point were significantly below 0 and numerically lowest for pairmates with the highest pre-IP similarity (4th quartile comparison to 0: \$t_{30} = -2.54\$, \$p = 0.017\$, \$d = 0.46\$, \$CI = [-0.023 \pm 0.019]\$ ); the pattern in CA1 was qualitatively opposite. Collectively, these results provide novel, theory-consistent evidence that remapping of competing representations in CA3/dentate gyrus is actively triggered by initial representational overlap.”

Reviewer 3, Comment 6. Although I suspect that this is not terribly likely to have biased the data, did the authors look for potential systematic effects of scene or item memorability, or scene-item congruency in the data? That is, were some scenes or items particularly well remembered in this dataset? Is there any evidence that such an effect may have systematically influenced LR or the quartile distribution? Is there any evidence for memorability as a function of particular scene-item pairings? I do not mean to send the authors down an analytical ‘rabbit hole’ here, but it would be informative if these kinds of influences can be simply and easily ruled out.

The Reviewer raises an interesting question about the degree to which there were differences across pairs of scenes as reflected in memorability and/or fMRI measures. In selecting stimulus pairs, we made an effort to minimize variability across pairs (or to at least avoid pairs that were at either extreme of the difficulty spectrum). However, there was nonetheless some variability across the scene pairs in terms of subjects’ performance on the associative memory tests. We have generated a new Supplementary Figure 1, copied below, which shows the mean associative memory accuracy by round for each pair of stimuli, rank ordered from highest mean accuracy (top of the left column) to lowest (bottom of the right column). As can be seen, all pairs were associated with a clear, robust improvement

across learning rounds. However, an ANOVA with factors of stimulus pair and round revealed a significant main effect of stimulus pair ($F_{17,510} = 4.08, p < 0.001, \eta^2 = 0.08$) as well as a significant interaction between stimulus pair and round ($F_{17,510} = 2.18, p = 0.004, \eta^2 = 0.02$). In other words, accuracy—and the rate of change in accuracy—varied across stimulus pairs. This is not particularly surprising nor do we view it as problematic that there would be differences across scene pairs. From our perspective, any such differences could be a meaningful source of variance as opposed to a bias or artifact. While detailed consideration of this point is beyond the scope of the current paper, including these data in the Supplement will allow for qualitative consideration of variance across stimulus pairs. In the new figure, we also include mean CA3/dentate gyrus pairmate similarity scores, by timepoint, for each set of scene pairmates. An ANOVA with factors of timepoint and stimulus pair did not reveal a significant main effect of stimulus pair ($F_{17,510} = 0.74, p = 0.760, \eta^2 = 0.01$) or an interaction between stimulus pair and timepoint in CA3/dentate gyrus ($F_{17,510} = 1.49, p = 0.093, \eta^2 = 0.02$). Thus, at least in terms of overall similarity values and experience-dependent changes in similarity, we did not see statistically significant differences across scene pairmates in CA3/dentate gyrus. Again, it is worth noting that we will make all of these data publicly available.

Supplementary Figure 1. Behavioral accuracy and CA3/dentate gyrus pairmate similarity scores for each set of scene pairmates. Scene pairmates are rank-ordered from highest (top, left column) to lowest (bottom, right column) mean accuracy on the associative memory test rounds. Individual plots of ‘behavior’ show mean associative memory test accuracy by learning round (1-6) for each set of scene pairmates. Mean test accuracy

across all rounds is denoted by the red dashed line. An ANOVA with factors of stimulus pair and round revealed a significant main effect of stimulus pair ($F_{17,510} = 4.08$, $p < 0.001$, $\eta^2 = 0.08$) as well as a significant interaction between stimulus pair and round ($F_{17,510} = 2.18$, $p = 0.004$, $\eta^2 = 0.02$). Individual plots of 'CA3/DG' show mean pairmate similarity scores in CA3/dentate gyrus by timepoint (1-5) for each set of scene pairmates. Each timepoint reflects similarity in CA3/dentate gyrus across successive learning rounds (1-2, 2-3, etc.). An ANOVA with factors of timepoint and stimulus pair did not reveal a significant main effect of stimulus pair ($F_{17,510} = 0.74$, $p = 0.760$, $\eta^2 = 0.01$) or an interaction between stimulus pair and timepoint in ($F_{17,510} = 1.49$, $p = 0.093$, $\eta^2 = 0.02$). Note: error bars reflect S.E.M.

Reviewer 3, Comment 7. In Figure 1e, I am unclear on why the data are being displayed as a cumulative number of pairs learned across the entire sample rather than averaged across the sample. It seems to me that the latter would be more informative.

We agree. We have modified the figure panel as shown below.

Reviewer 3, Comment 8. This is very minor and a matter of personal preference, but I actually think that the readability of the paper could be somewhat improved by not abbreviating with 'LR' and 'IP'. On more than one occasion, I had to go back and remind myself what those acronyms stood for since they are very specific to this paradigm and not terms used by the field more broadly. Simply spelling out these terms – at least in the text, if not the figures – could help readers stay oriented, particularly given that they are not lengthy terms.

We thank the Reviewer for the suggestion. We have revised the text to minimize the use of these abbreviations. We do still use the abbreviations in figures, but we have made sure to define the abbreviations in the figure captions. Additionally, we retained the abbreviations in a few places in the main text when referring to “pre-IP” and “LR-1” simply because we felt this was easier to read than “pre-inflection point” and “learning round – 1.” However, we have made sure that in all places where “pre-IP” and “LR-1” are used, there is a nearby reminder of what these abbreviations mean.

References.

1. Molitor, R. J., Sherrill, K. R., Morton, N. W., Miller, A. A. & Preston, A. R. Memory reactivation during learning simultaneously promotes dentate gyrus/CA2,3 pattern differentiation and CA1 memory integration. *J. Neurosci.* (2020) doi:10.1523/JNEUROSCI.0394-20.2020.
2. Schlichting, M. L., Mumford, J. A. & Preston, A. R. Learning-related representational changes reveal dissociable integration and separation signatures in the hippocampus and prefrontal cortex. *Nat. Commun.* **6**, 8151 (2015).
3. Ritvo, V. J. H., Turk-Browne, N. B. & Norman, K. A. Nonmonotonic Plasticity: How Memory Retrieval Drives Learning. *Trends Cogn. Sci.* **23**, 726–742 (2019).
4. Newman, E. L. & Norman, K. A. Moderate Excitation Leads to Weakening of Perceptual Representations. *Cereb. Cortex* **20**, 2760–2770 (2010).
5. Rouhani, N. & Niv, Y. Signed and unsigned reward prediction errors dynamically enhance learning and memory. *eLife* **10**, e61077 (2021).
6. Kim, G., Lewis-Peacock, J. A., Norman, K. A. & Turk-Browne, N. B. Pruning of memories by context-based prediction error. *Proc. Natl. Acad. Sci.* **111**, 8997–9002 (2014).
7. Kim, G., Norman, K. A. & Turk-Browne, N. B. Neural Differentiation of Incorrectly Predicted Memories. *J. Neurosci.* **37**, 2022–2031 (2017).
8. DuBrow, S., Rouhani, N., Niv, Y. & Norman, K. A. Does mental context drift or shift? *Curr. Opin. Behav. Sci.* **17**, 141–146 (2017).

REVIEWERS' COMMENTS

Reviewer #1 (Remarks to the Author):

The authors have provided thorough and clear responses to my comments as well as the other reviewers'. The additional analyses are very helpful, and I appreciate the additional clarifications the authors have added. The new supplementary figures provide informative context and interesting data points.

I appreciate the analyses and plots the authors have included in the response letter, and I agree with the authors that they do not need to be included in the supplementary materials.

I have no further comments. This is a timely, well-designed study and set of analyses and a well-written paper which provides interesting new insights into the nature of integration and differentiation within the hippocampus.

Reviewer #2 (Remarks to the Author):

The authors have addressed my concerns and have provided additional information for the reader. I have no additional comments.

Reviewer #3 (Remarks to the Author):

The authors have done a commendable job of addressing my concerns, and in my opinion the concerns of the other reviewers as well. I think what was already a good paper has been substantially improved, and I thank the authors for their efforts. I support publication of the manuscript, and have no further comments.